# Reversible gene silencing through frameshift indels and frameshift scars provide adaptive plasticity for *Mycobacterium tuberculosis*

Aditi Gupta[1✉] & David Alland [1✉]

*Mycobacterium tuberculosis* can adapt to changing environments by non-heritable mechanisms. Frame-shifting insertions and deletions (indels) may also participate in adaptation through gene disruption, which could be reversed by secondary introduction of a frame-restoring indel. We present ScarTrek, a program that scans genomic data for indels, including those that together disrupt and restore a gene's reading frame, producing "frame-shift scars" suggestive of reversible gene inactivation. We use ScarTrek to analyze 5977 clinical *M. tuberculosis* isolates. We show that indel frequency inversely correlates with genomic linguistic complexity and varies with gene-position and gene-essentiality. Using ScarTrek, we detect 74 unique frame-shift scars in 48 genes, with a 3.74% population-level incidence of unique scar events. We find multiple scars in the ESX-1 gene cluster. Six scars show evidence of convergent evolution while the rest shared a common ancestor. Our results suggest that sequential indels are a mechanism for reversible gene silencing and adaptation in *M. tuberculosis*.

[1] Center for Emerging Pathogens, New Jersey Medical School, Rutgers University, Newark, NJ, USA. ✉email: aditi9783@gmail.com; david.alland@rutgers.edu

Mycobacterium tuberculosis must adapt to a wide range of conditions from aerosol droplets to intracellular and extracellular environments, and from replicative states to latency[1–7]. Rapid adaptation is often explained by epigenetic events including activation of the stringent response, the *dosR* regulon, and a variety of other reversible transcriptional or metabolic changes that improve survival in response to specific environmental stresses[5,8–12]. Small insertions and deletions (indels <50 bp) can have major effects on gene function by introducing frameshifts or stop codons. The functional effects of small indels have been poorly characterized despite evidence of their frequent occurrence[13,14], and their role in microevolution and host adaptation remains underexplored.

Two recent reports demonstrated that *M. tuberculosis* can achieve a diminished rate of growth and broad tolerance to antibiotics through frame-shifting indels in the *M. tuberculosis* *glpK* gene[15,16]. Furthermore, these effects were demonstrated to be rapidly reversible when a second indel restored the frame and thus the function of the mutated gene[15]. This sequential accumulation of two frame-shifting indels in the same gene provides a potential new mechanism by which pathogenic bacteria can reversibly alter their phenotype in response to drug exposure or host infection. To investigate the more general prevalence of this phenomenon in *M. tuberculosis* outside of the *glpK* gene, we developed and applied two new computational tools to analyze the sequence complexity of the bacterial genome and to detect sequential frame-shifting and restoring indels in 5977 recently published *M. tuberculosis* clinical genomes[17–19]. Using these tools, we analyzed the repetitive sequence structure (using linguistic complexity) and the non-uniform base distribution such as the presence of homopolymers (using Shannon's entropy) in the reference *M. tuberculosis* genome and explored the relationships between indel incidence and sequence composition. We found "frameshift scar" remnants of frame-shifting and restoring indels in a relatively large number of *M. tuberculosis* isolates. Here we describe the results of this investigation and provide in silico evidence that the sequential appearance of indels may facilitate the adaptation of *M. tuberculosis* to dynamic environments.

## Results

**Indels cause genomic plasticity in *M. tuberculosis*.** *M. tuberculosis* strain H37Rv has 4111 genes that account for 91.3% of its genome. Of these, 461 genes are deemed essential for in vitro growth by saturating transposon mutagenesis[20] and 165 genes belong to the repeat-rich PE-PPE gene family[21], two of which are essential[20]. We developed a bioinformatics tool called ScarTrek that detects indels and indel scars using restrictive filters (see Methods). We compared the indel detection accuracy of ScarTrek to SAMtools and GATK HaplotypeCaller on a "gold-standard" test set where pre-defined indels were artificially introduced into 10 *M. tuberculosis* genomes. ScarTrek's indel detection was superior to SAMtools and was comparable to GATK HaplotypeCaller (Table S1 and Figs. S1 and S2). Further testing on 402 clinical *M. tuberculosis* genomes revealed high concordance between indels detected by ScarTrek and other tools: 97.6% of indels detected by ScarTrek were also detected by either SAMtools or GATK HaplotypeCaller or both (Fig. S2a). At low read coverages, ScarTrek detects more indels correctly, and has performance comparable to GATK HaplotypeCaller at higher coverage (Table S1 and Fig. S1). ScarTrek was then used on the genomes of 5977 clinical *M. tuberculosis* isolates to study the contribution of indels in the plasticity of the *M. tuberculosis* genome. We found 16,693 unique indels of which 13,692 (82%) were in sequences annotated as genes and 3,001 (18%) were in intergenic regions. The density of indels was higher in intergenic regions, with one indel occurring in every 17,937 bases on average in a given isolate, compared to one indel occurring in every 74,497 bases in genic regions. The 5976 isolates that had at least one genic and one intergenic indel had significantly different indel densities in intergenic and genic regions (two-tailed *p* value = 0.0, Welch *t* test statistic: 63.97). This skewed indel distribution may be due to a lower fitness cost for indels in intergenic regions. In keeping with this observation, we expected that in-frame indels (indels occurring in multiples of three nucleotides) would be more frequent than frameshift indels in genic regions. However, unique frameshift indels in our entire dataset of 5977 isolates (13,521 indels, 81%) were far more abundant than the in-frame indels (3172 indels, 19%) primarily because the most common indels were one nucleotide in length (Fig. 1a). Indels were plentiful in PE-PPE genes, although the low sequence complexity of these genes can lead to more errors in indel-calling, decreasing the precision of this estimate. PE-PPE genes contained 12.03% of all indels, and 95.7% (158 of 165) of these genes contained at least one indel. Indels were also more frequent in non-PE-PPE genes that were classified as nonessential, with indels present in 77.1% of such genes (2621 out of 3399) compared to only 28.4% (131 out of 461) of essential genes (Fig. 1b, Table S2). Although indels were seen in one-third of essential genes, 90.8% of these genes had only one or two indels in the 5977 isolates (Fig. 1b). Nine isolates had indels in the ribosomal RNA (*rrs*) gene, which raised the possibility of contamination or mixed infections. Metagenomics analysis revealed that all of these 9 isolates mapped to *M. tuberculosis* and did not show evidence of mixed infection or contamination (Table S3).

We examined the intra-genic distribution of indels to better understand their fitness cost. Focusing on the 4018 protein-coding genes only, we found that, for essential genes, the majority of indels were at the gene termini (Fig. 1c), a trend observed in protein-coding genes in humans[22–24]. Indels at gene termini are suggested to be neutral: those occurring at C-terminus may have little or no impact on the translated protein while those at the N-terminus may be rescued by an alternate start codon downstream of the indel[22]. Eleven of the 24 frameshift indels in the first one-tenth of essential genes were in the region preceding the first occurrence of the ATG start codon, and several other indels were close to a potential alternate ATG translation start site, especially since the predicted gene start sites do not always agree with experimentally validated start sites[25] (Table S4). In contrast, indels were uniformly distributed along the gene length for the PE-PPE and nonessential genes (Fig. 1c), suggesting that indels in these genes develop under neutral or positive selective pressure.

**Indels are associated with low complexity genomic sequences.** The high incidence of indels, particularly frameshift indels, in the *M. tuberculosis* genome suggests that indels might serve as a mechanism for adaptive gene variation. To explore the influence of sequence composition on indel frequency, and thus genome plasticity, we developed "FindingInfo", a computational tool to compute linguistic complexity and Shannon's entropy along the length of a nucleotide sequence. The linguistic complexity (*LC*) score considers the ordering of nucleotides and quantifies repetitiveness in a sequence[26]. Shannon's entropy (*H*) quantifies nucleotide diversity in a sequence and is calculated from nucleotide frequencies without considering their order[27,28], identifying stretches of homopolymers and other regions with low nucleotide diversity. These scores lie between 0 and 1 with low scores indicating the presence of repeats and non-uniform nucleotide composition in a sequence, i.e., regions of low

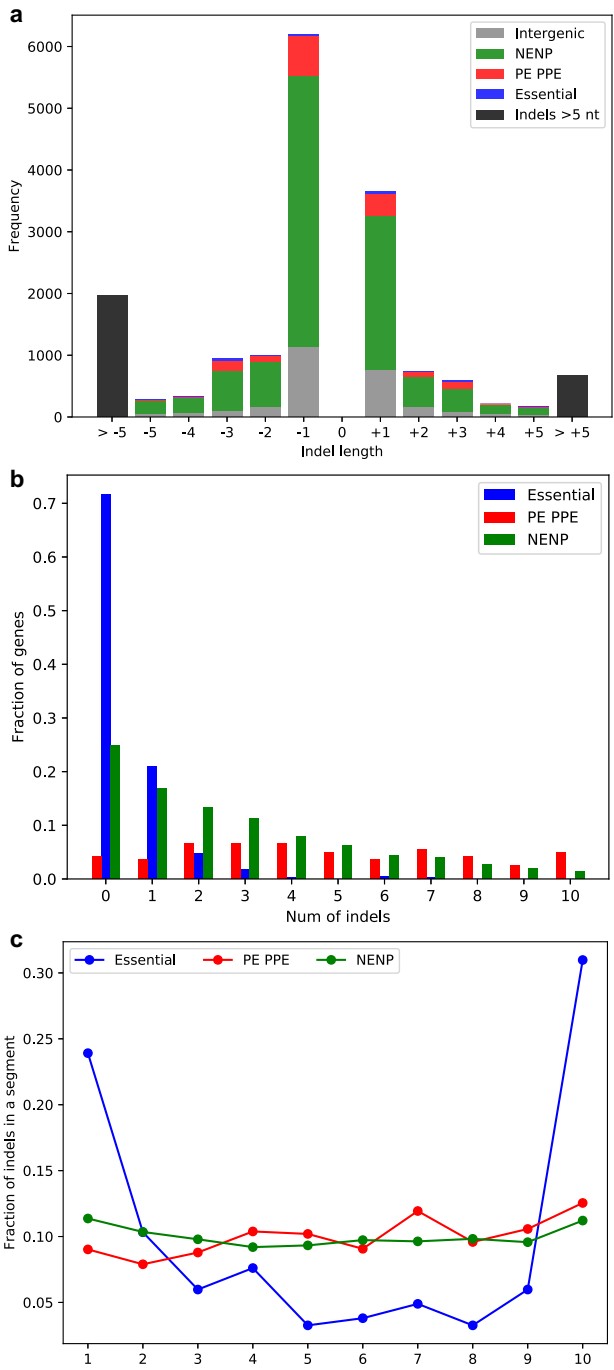

**Fig. 1 Properties of unique insertions and deletions (indels) identified in 5977 clinical isolates of *M. tuberculosis*. a** Indel frequency (*y*-axis) is shown as a function of indel size (*x*-axis) for indels found in the intergenic regions (gray), nonessential non-PE-PPE genes (NENP, green), PE-PPE genes (red), and essential genes (blue). Positive indel sizes (*x*-axis) denote insertions and negative indel sizes denote deletions. The majority of indels are one nucleotide long and larger indels are less frequent (indels >5 nt are shown in black, see Table S10 for size distribution of >5 nt long indels). **b** Occurrence of unique indels (*x*-axis) is shown for essential (blue), PE-PPE (red), and the remaining non-essential non-PE-PPE (NENP, green) genes (data in Table S2). Approximately 70% of the essential genes have no indels whereas the highly repetitive PE-PPE genes have an almost uniform distribution of the number of unique indels that are found in these genes. **c** Fraction of unique indels in essential, PE-PPE, or NENP genes (*y*-axis) are shown along the gene length represented as gene-segments (*x*-axis). Each gene was divided into ten equal segments. The majority of indels in essential genes were found in the first and the last gene segments, unlike the PE-PPE and the remaining genes where indels were uniformly distributed throughout the gene.

sequence complexity. We used FindingInfo to compute these complexity measures for each position in the *M. tuberculosis* H37Rv reference genome (GenBank ID: AL123456.3). As shown previously in a number of other organisms[29,30], we found that *M. tuberculosis* indels were significantly enriched in regions of low sequence complexity, with the mean *LC* of 16,693 indel and 16,385 randomly sampled (without replacement, Table S5) non-indel positions being 0.541 versus 0.59, respectively (Welch *t* test statistic: −24.71, two-tailed *p* value: 1.45e-133), and the mean *H* of indel and non-indel positions being 0.928 versus 0.94, respectively (Welch *t* test statistic: −23.21, two-tailed *p* value: 3.37e-118). Moreover, we noted that complexity scores formed "indel pockets" with scores declining from 7 bases upstream of an indel to 14 bases downstream of the indel position in orphan

indels (i.e., indels that were at least 100 bases away from other indels) (Figs. 2a and S3, Supplementary Datasets 1–4).

We found that almost all genes had intermittent regions of low-sequence complexity that could serve as indel pockets, regardless of their overall complexity. Thus, almost all genes have regions with an increased capacity for genetic variation via indels (Fig. 2b and c). Even the essential gene with the highest *LC* and *H* scores (*Rv3902c*) had regions of low complexity interspersed along the length of the gene (Fig. S4). It is noteworthy that the mean complexity of PE-PPE genes was inversely correlated with the number of indels in the genes, which suggests that low sequence complexity leads to more indels although increased error rates for indel detection in these highly repetitive genes is an alternate explanation (Fig. S5, Pearson correlation coefficient and two-tailed *p* value for *LC* scores: −0.402 and 8.5e-8, for *H* scores: −0.35, 3.96e-6). This trend did not hold true for either essential genes (Pearson correlation coefficient and two-tailed *p* values for *LC* scores: −0.047 and 0.312, for *H* scores: −0.027 and 0.57) or the remaining genes that were neither essential nor PE-PPE (Pearson correlation coefficient and two-tailed p values for *LC* scores: −0.055 and 0.001, for *H* scores: -0.073 and 1.4e-5). Because low complexity regions are interspersed throughout the length of a gene, we computed the fraction of the gene length that had complexity scores lower than threshold scores of 0.551 for *LC* and 0.932 for *H*, as defined by the lowest complexity scores in the indel pockets of orphan indels seen in Fig. 2a. We found that the low-complexity fraction of PE-PPE genes was positively correlated with the number of unique indels (Pearson correlation coefficient and two-tailed *p* value for *LC* scores: 0.38 and 4.2e-7, Fig. 2b, for *H* scores: 0.31, 6.06e-5, Fig. 2c). In contrast, the low-complexity gene fraction of essential genes and non-essential non-PPE genes was uncorrelated with the number of unique indels in these genes (Pearson correlation coefficients for *LC* and *H* scores for essential genes: 0.028 and 0.035, respective two-tailed p values: 0.55 and 0.45; Pearson correlation coefficients for *LC* and *H* scores for nonessential non-PPE genes: 0.053 and 0.061, respective two-tailed *p* values: 0.002 and 0.0003). Moreover, the number of unique indels was strongly correlated with gene length for PE-PPE genes (Fig. 3, Pearson correlation coefficient and two-tailed p value: 0.877 and 7.69e-54), moderately correlated for non-essential non-PPE genes (Pearson correlation coefficient and two-tailed *p* value: 0.464 and 1.12e-40), and weakly correlated for essential genes (Pearson correlation coefficient and two-tailed *p* value: 0.222 and 1.51e-6). Thus, the abundance of low sequence

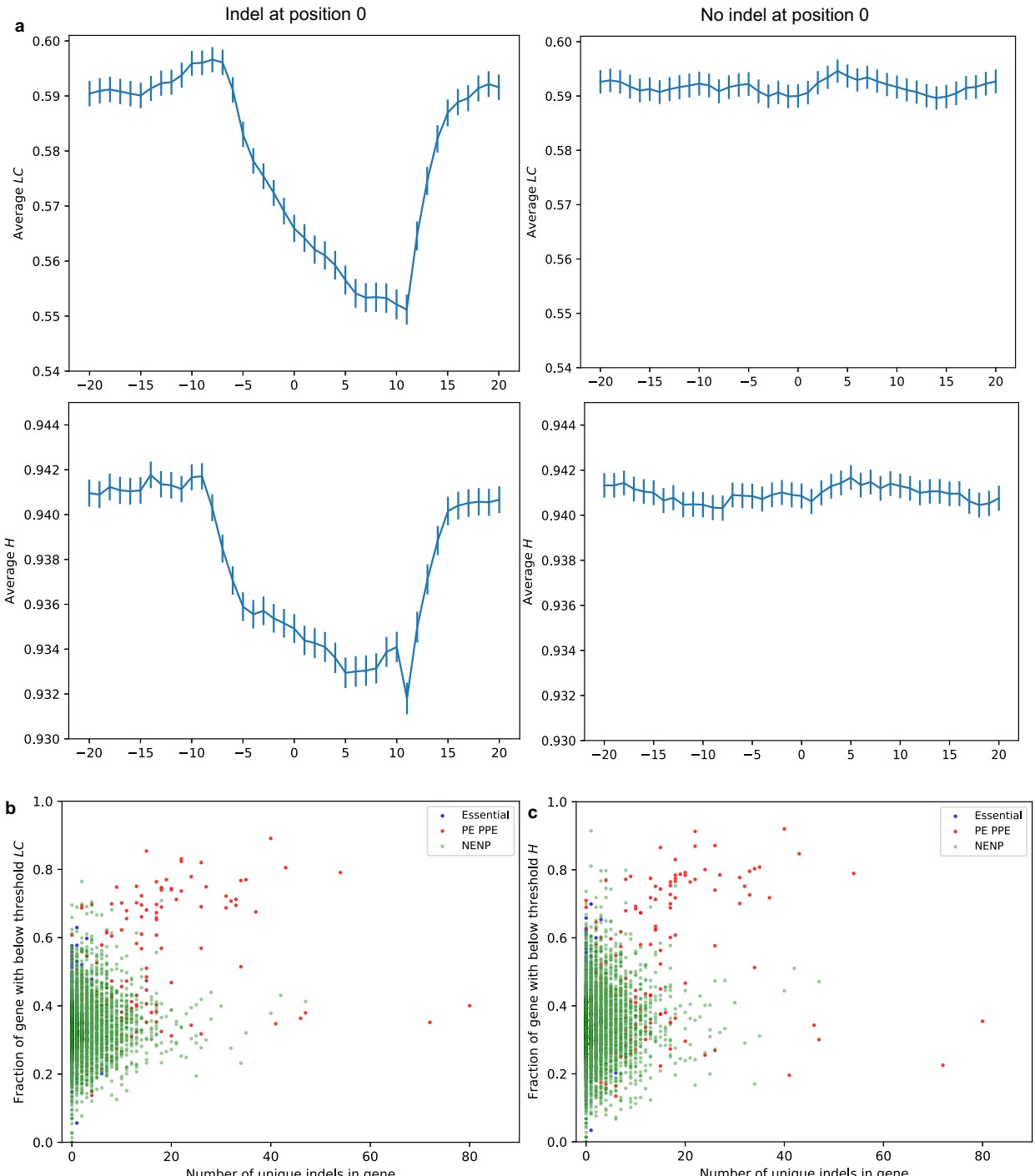

**Fig. 2 Relationship between the frequency of indel occurrence and sequence complexity in the *M. tuberculosis* genome. a** The linguistic complexity (*LC*) and Shannon's entropy (*H*) scores are shown for 20 positions upstream and downstream of orphan indels that do not have another indel within 100 bases (*n* = 5172), and randomly sampled non-indel sites that do not have an indel within 100 bases of the site (*n* = 5775). The sequence complexity profile in the vicinity of the indel and non-indel sites are different. The average *LC* and *H* scores adjacent to the indel sites show a decrease in the complexity scores 7–10 bases before the indel position (indicated by 0 on the *x*-axis) and 15–18 bases after the indel position. The error bars represent ±1 SEM (standard error of the mean). See Fig. S3 for density distributions of *LC* and *H* scores at and around these indel and non-indel positions. **b** The fraction of each gene that has a complexity score below the threshold score for *LC*, and **c** the fraction of each gene that has a complexity score below the threshold score for *H* in essential (blue), PE-PPE (red), and the remaining genes that are neither essential nor PE-PPE (NENP, faded green). The threshold scores (*H* = 0.932, *LC* = 0.551) are the lowest *H/LC* scores in the indel pockets shown in Fig. 2a.

complexity regions in the PE-PPE genes likely contributes to the high indel incidence in these genes (and possibly high indel error rate compared to high complexity regions), but indels may be restricted in non-PPE genes by negative selection.

**Gene scarring**. Indels that disrupt the genic reading frame can cause a loss of gene function due to truncated transcription and alterations in the protein coding sequence. This deleterious effect of a frameshift indel can, in theory, be overcome if a second

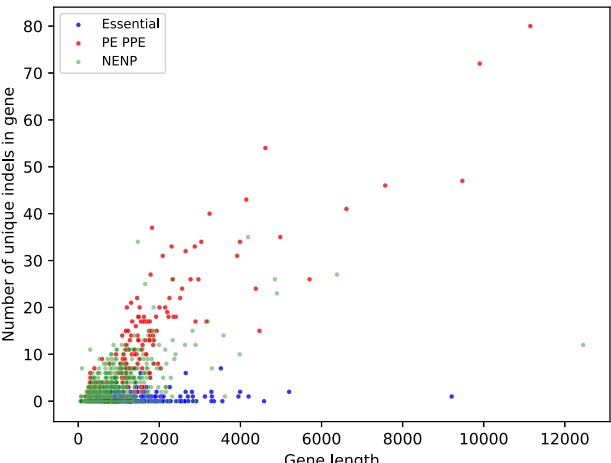

**Fig. 3 Indel incidence and gene length.** The number of unique indels found in PE-PPE genes (red), in essential genes (blue) and in a random sample of 500 non-essential and non-PE-PPE genes (NENP, faded green) is shown as a function of gene length (nt). The number of unique indels in PE-PPE genes strongly correlate with the gene length, while this correlation is moderate for nonessential and non-PE-PPE genes, and weak for essential genes.

frameshift indel restores the reading frame. Safi et al.[15] showed that a second frame-shifting indel in the *glpK* gene of *M. tuberculosis* restored wild-type growth and reversed a drug-tolerant phenotype in a slow-growing mutant that had another frameshift indel in the gene. To determine if a sequential appearance of frame-shifting and restoring indels is a common mechanism of adaptive evolution, we used our ScarTrek program to identify sets of gene disrupting and restoring indels from whole-genome sequencing data of *M. tuberculosis* isolates. On the premise that the first frameshift indel "wounds" the transcript (wounding indel) and the second frame-restoring indel fixes it (fixing indel), we call the recovered sequence "scarred" because the two indels may be far enough to alter one or more residues in the final gene product. Thus, a "frameshift scar" describes a set of wounding and fixing indels such that they disrupt the reading frame individually but not when present together. The ScarTrek algorithm identifies scars by introducing each frame-shifting indel present in a gene one at a time and then all together to determine if the transcript was disrupted by one (or more) indels and then restored when all indels are present. ScarTrek analysis of WGS data from 5977 clinical isolates of *M. tuberculosis* identified 402 isolates that had 74 unique scars in 48 non-PE-PPE genes (Fig. 4 and Table S6). Each of the 155 indels in these scars was confirmed by manual inspection of sequencing reads aligned to the reference genome using Tablet[31]. Although only 402 unique isolates (6.72%) had scarred genes, 5957 (99.65%) of the isolates had at least one of the 140 unique scar indels. This is primarily due to a third of the scar genes where the wounding indel alone is present in a large number of isolates (Table S7). None of the scarred genes were deemed essential for in vitro growth by earlier studies (Table S8)[20]. Several of the scarred genes were in the ESX-1 secretion system (*eccB1*, *eccCa1*, *esxA*, *espI*, *eccD1*, *espB*, and *eccE1*, see Fig. 4), with five of these genes having multiple unique scars (*eccB1*, *eccCa1*, *espI*, *eccD1*, and *eccE1*). The ESX-1 secretion system is important for virulence and is critical in eliciting the immune response to the *M. tuberculosis* infection[32–34].

The location of frameshift indels within a gene has consequences on gene function. If the first i.e., wounding frameshift indel occurs such that it renders the gene-product nonfunctional, there is a high selection pressure to restore the reading frame via a second frameshift indel. However, if the distance between the frame-shifting and restoring indel is large, the recovered gene-product will differ from the wild-type protein and may alter the protein function. We found that the indels constituting a scar often occurred close to each other, limiting the number of residues that change in the restored protein (Fig. 5a). We divided each gene into ten equal segments and determined the gene segments in which the frameshift scars occurred. Sixty of the 69 scars (87%) that were formed by exactly two indels (for straightforward analysis of distances between scar indels) had scar indels co-located in either the same or the adjacent gene segment (Fig. 5b). This suggests that in most cases, a frameshift scar does not substantially alter the gene product. We also compared the fraction of the gene that was low complexity in scarred versus non-scarred genes that are non-PE-PPE and non-essential for in vitro growth (note that non-essentiality has not been tested in a human host). This analysis showed that the low-complexity gene fraction was similar in both groups (two-tailed *p* values for comparing the means for low *LC* gene-fraction and low *H* gene-fraction: 0.055 and 0.061, respective Welch *t*-test statistics: 1.97 and 1.92, Fig. 5c–d). Thus, while overall gene complexity influences indel occurrence, it may not determine whether scars form in certain genes. This suggests that the pool of genes that can contribute to adaptation through wounding and scarring is potentially large, and the biological effect of a frameshift wounding indel is likely to be the most important predictor of whether a particular gene acquires a frameshift scar.

**Convergent evolution of frameshift scars.** To investigate whether the occurrence of identical frameshift scars in different clinical isolates is due to a common ancestor or convergent evolution, we generated a Bayesian tree of the 57 isolates that had scars in the *espI* gene with 200 isolates randomly selected without replacement from a set of phylogenetically distant isolates (Fig. 6a). While most of the isolates that had identical frameshift scars were nearby in the phylogenetic tree, suggesting a common ancestor (for example: Fig. 6a, scar F shown in gray), one identical scar (Fig. 6a, scar C shown in green) was present in two isolates that were far apart in the tree. Each of the two indels that constitute this scar C appeared independently in the phylogenetic tree, suggesting convergent evolution for this scar (Fig. S6a). In contrast, indels that constitute scar F (gray) in *espI* gene were present in a single clade in the tree (Fig S6b). Additionally, several scars shared one or more indels with other scars (Fig. 6b: scars B, F, and I share two indels; scars A, C, I, and J share one indel; scars B and G share one indel; and scars C and G share one indel). This suggests that convergent evolution, as well as direct descent of scar indels, led to multiple scars in *espI*. A similar phylogenetic analysis of all frameshift scars suggested that six scars in five genes developed independently (Fig. S7b, one scar each in *Rv2542*, *sppA*, *sigM*, *espI*, and two scars in *pks12*). Further, convergent evolution of some scar indels in four genes (*sigM*, *aofH*, *Rv2561*, and *espI*) was observed (Fig. S7c). For example, disruption of the *Rv2561* gene by one indel was restored by five distinct indels (Fig. S7c). This convergent evolution of scars suggests that the scar indels did not appear simultaneously. Counting each scar clade as a separate "scar occurrence" (see Methods, in brief: for each scarred gene, a phylogenetic tree was constructed from isolates that had a scar in that gene along with 200 phylogenetically distant isolates, and total number of scar clades from all the trees are reported), we noted an average of 223.3 evolutionarily independent scar occurrences in our sample of 5977 isolates resulting in an incidence rate of 3.74% (number of scar occurrences in 10 independent phylogenetic constructs: 208, 236, 225, 241, 224, 238, 214, 216, 218, 213; average: 223.3, standard

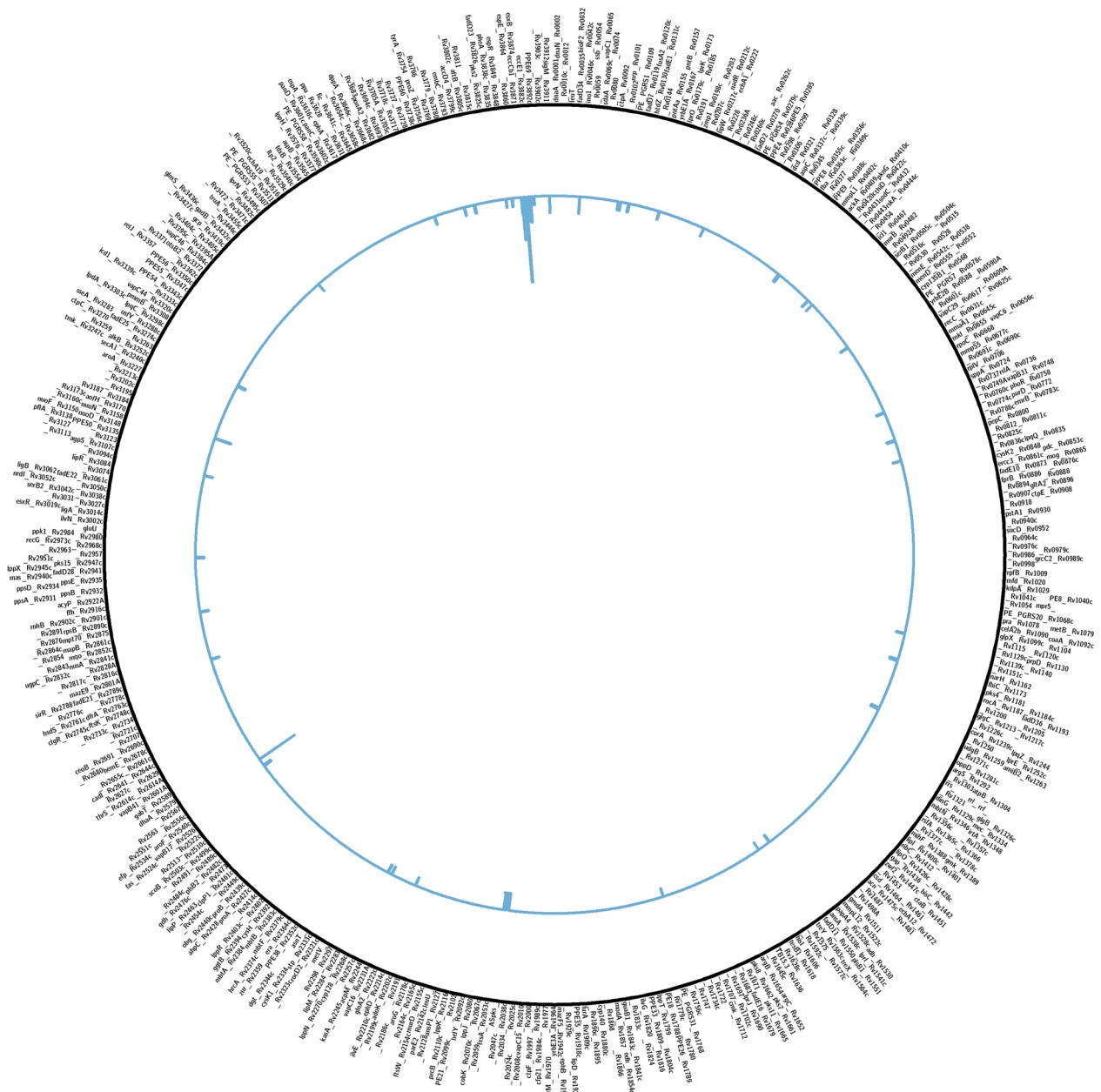

**Fig. 4 Genome plot of *M. tuberculosis* H37Rv showing distribution of unique scars.** The histogram shows number of unique scars found in the 48 scarred genes. Note the high incidence of scars in the ESX-1 gene cluster.

deviation: 10.96). However, scars in 15 genes were found only in close evolutionary relatives (Fig. S7d, genes are *cut1*, *eccB1*, *eccCa1*, *eccD1*, *espB*, *fusA2*, *ltp1*, *Rv0045c*, *Rv0176*, *Rv0458*, *Rv1132*, *Rv1575*, *Rv2216*, *Rv2561*, and *Rv0823c*), suggesting that direct descent of scars from an evolutionary ancestor is common.

Although homoplasies in individual indels were three times more common in indels found in low complexity regions (Fig. S8), indicating that the local sequence composition (and not selection pressure) may lead to such independent indel occurrences, scars are composed of multiple indels; and thus, the independent occurrence of identical scar sequences in evolutionarily distant isolates is likely due to selection pressures. The *espI* gene contained the greatest number of scars over the entire set of *M. tuberculosis* genomes analyzed, with 10 unique scars in 57 different clinical isolates (Fig. 6). The *espI* gene is part of the ESX-1 gene cluster and is implicated in negative regulation of the ESX-1 secretion system when the cellular ATP levels are low[35].

Because of the high abundance of scars in this gene and in the ESX-1 gene cluster, it is possible that indels have a particularly important role in regulating this cluster. Thus, wound and scar formation may be linked to *M. tuberculosis* pathogenesis or immune evasion in addition to their established role in reversible drug tolerance.

## Discussion

Little is understood about the fitness effects of indels, their distribution in the genome, and their role in adaptive evolution. Indels may provide a form of genomic plasticity that enables quick and readily reversible adaptation to changing environments. Indels can substantially affect gene function by disrupting the reading frame of a gene. However, these types of genomic perturbations can also result in high fitness costs. Our analysis of WGS data from 5977 clinical *M. tuberculosis* isolates reveals that the tradeoff between evolvability and maintaining viability may be achieved, in part via a

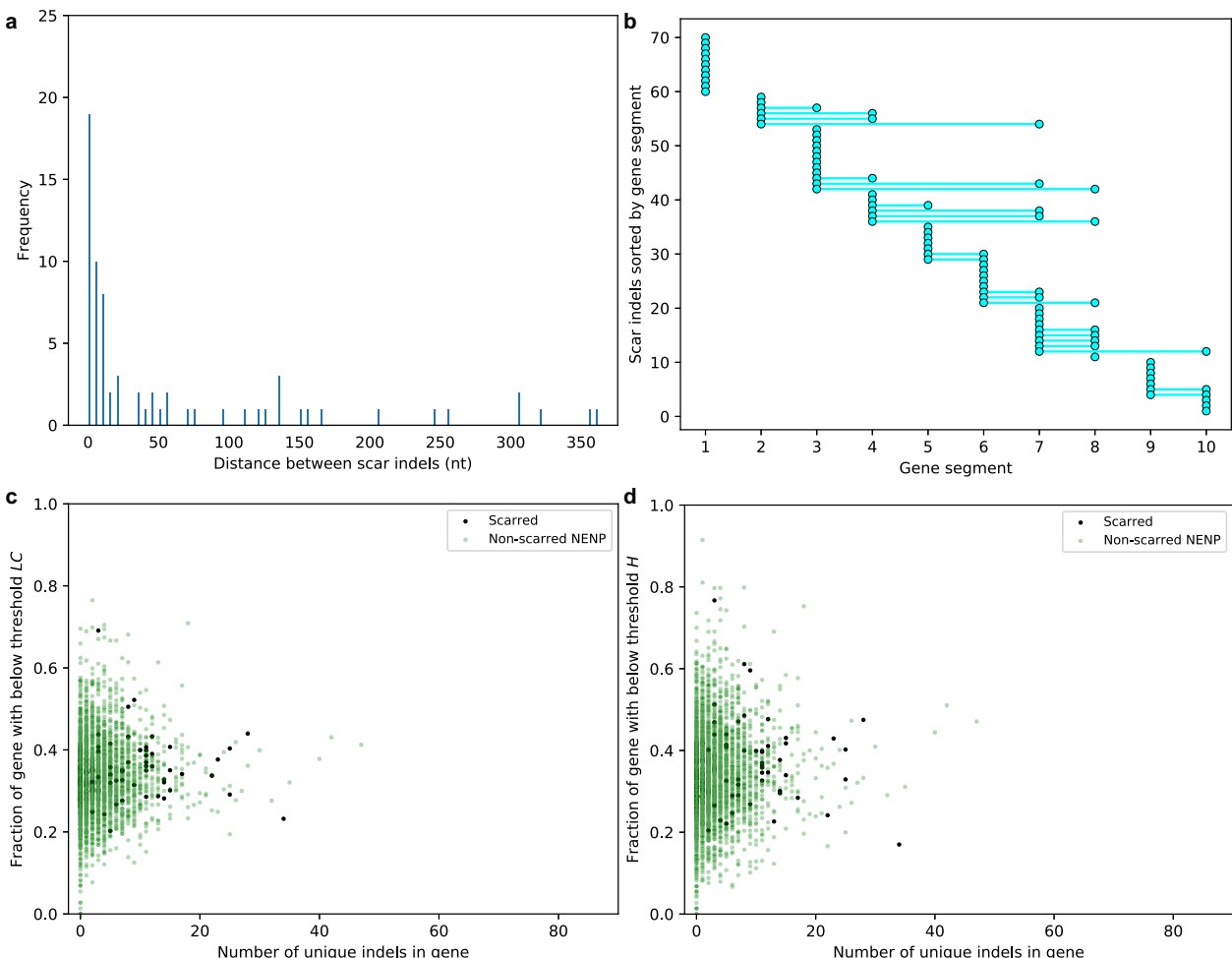

**Fig. 5 Indel properties of frameshift scarred genes. a** The distribution of distances between indels that together form a scar. In the 69 scars that resulted from exactly two indels, the majority had a distance of <10 nucleotides between the two indels. **b** Gene segments are shown for indels in scars that had exactly two indels. Each gene is divided in ten equal segments to show the locations of scar indels in the genes. Each indel is shown as a filled cyan circle and the two scar indels are connected by a cyan line if they occur in different gene segments. The absence of a cyan line indicates that both indels occurred in the same gene segment. 86% of these 69 frameshift scars had both the frame-shifting and frame-restoring indel in the same or neighboring gene segment. **c** and **d** The gene-fraction below threshold complexity scores is shown for scarred genes (black filled circles) and the non-essential and non-PE-PPE genes that do not have scars (non-scarred NENP, faded green circles), with **c** showing data for *LC* and **d** showing data for *H*. The threshold scores (*H* = 0.932, *LC*=0.551) are the lowest *H/LC* scores in the indel pockets in Fig. 2a.

non-uniform distribution of indels in the bacterial genome. We find high indel incidence in the intergenic and repetitive PE-PPE genes and an intermediate incidence in non-essential, non-PE-PPE genes. Genes essential for in vitro growth have the lowest incidence of indels. The functional consequences of indel abundance in these four genomic categories may also be different. Indels in intergenic sequences may affect gene expression, but will likely have a lower fitness cost because they do not disrupt protein sequences. Indels in the PE-PPE genes may provide flexibility for encoding different antigens and evading the host immune response[36,37]. Finally, indels in non-essential non-PE-PPE and in some essential genes may regulate gene function through an on-off mechanism by successively introducing and then resolving frameshifts via wounding indels and fixing indels. Understandably, essential genes have the lowest frequency of indels due to the high fitness cost of indels in those genes.

Analyzes of indels that have already occurred in the *M. tuberculosis* genome, while conveying some information about the role of indels in adaptive evolution, says nothing about the evolutionary potential encoded in the bacterial genome. We

utilized the linguistic complexity and Shannon's entropy measures to detect regions of low sequence complexity (repetitive sequences and homopolymer regions) and found that low sequence complexity is positively correlated with indel occurrence. In addition, low complexity regions were interspersed throughout all genes with the highly mutable PE-PPE genes containing a larger proportion of low complexity regions. This abundance of low-complexity regions highlights the plasticity of *M. tuberculosis* genome.

There are two main approaches for detecting indels from sequencing data: de novo assembly and alignment to a reference genome. It can be difficult to identify indels using either approach because indels usually occur in low complexity sequences that may be prone to errors in genome assemblies and read alignments[38,39]. Long homopolymer runs also introduce errors during the PCR step[39,40]. While different variant callers have a high concordance in SNP calling, the same is not true for indel calling due to a high number of false positives[41,42]. Assembly-based approaches are well suited for detecting large indels that are very likely to be missed by the alignment-based approaches[39,43].

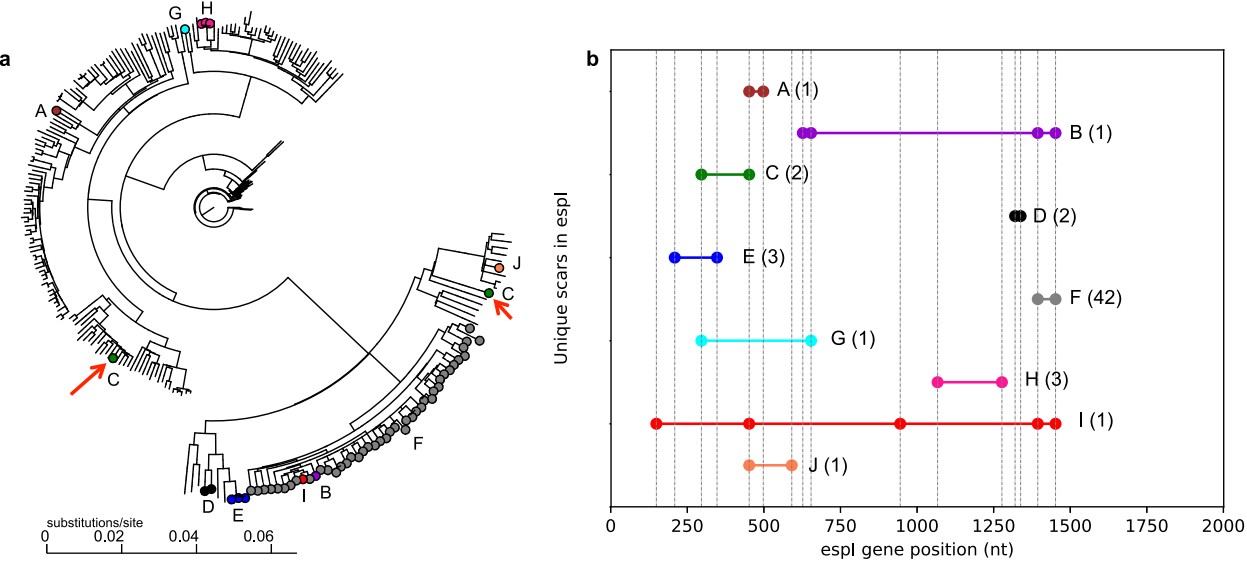

**Fig. 6 Unique frameshift scars in the *espI* gene. a** A Bayesian tree of the 57 isolates that had scars in the *espI* gene and 200 randomly selected phylogenetically distant isolates is shown. The ten unique scars are identified by letters A–J as well as different colors. Absence of a colored node indicates a randomly selected independent isolate without a scar in the *espI* gene. Identical scars indicated in green nodes (scar C, red arrows) appeared independently in phylogenetically distant isolates, suggesting convergent evolution. Although unrooted, the Newick tree visualizations are implicitly rooted at the top node. **b** The indels (filled circles) in the ten unique scars (each identified by a different color and letter) seen in our dataset of 5977 isolates are shown along the length of the *espI* gene. The number of isolates that have a given scar is indicated in parenthesis. The gene positions where scar indels occurred are shown by vertical dashed gray lines.

However, direct comparison of indel-calling in human genomes by assembly-based and alignment-based approaches showed that alignment-based approaches achieved a much higher precision and recall[44]. Recent indel callers combine both approaches by re-assembling the indel region after an initial read-alignment[45–47]. Even though indel calling remains problematic and downstream filtering of variants is needed to improve accuracy[39,40], ScarTrek detects a lower number of false positives and has high concordance with other indel callers by focusing on indels with strong read support. Our finding that indel frequency per sample shows a tri-modal distribution according to *M. tuberculosis* lineage (Fig. S14), with the "modern" lineages 2, 3, and 4 showing fewer indels than the "ancient" lineages 1 and 5 provides further validation for the accuracy of our indel calls.

We found frameshift scars in 48 *M. tuberculosis* genes in 402 *M. tuberculosis* isolates, with multiple scars detected in the ESX-1 secretion system. It is likely that ScarTrek underreports scars because it looks for them after all the scar indels have occurred. If two indels occur such that a frameshift is not detected (for example, insertion of a nucleotide followed by deletion of a nucleotide at the same position, or insertion of one nucleotide followed by insertion of two nucleotides adjacent to the first insertion), then the algorithm will ignore them. Indeed such events are described in Safi et al[15]. Thus, the total number of scars in the *M. tuberculosis* genome is likely higher than reported here. There is no adaptive benefit for a gene to be both wounded and fixed at the same time (i.e., scarred as a single event). Further, simultaneous development of multiple indels with the same genetic region is likely to be very low. Thus, we propose that most of the scars that we detected represent distinct, multi-stage, mutational events. Based on these data, we propose that indels may be a commonly used mechanism for transient adaptive evolution in *M. tuberculosis*.

## Methods
**ScarTrek: indel and scar identification.** The pseudocode of the ScarTrek software is shown in Fig. S9. Of the 6509 isolates in our dataset, 5977 had a read mapping

rate of 50% or higher AND had average genome-wide depth of 20 or higher (i.e., each position in the bacterial genome had, on average, 20 or more reads mapped to it, Fig. S10). Further, >95% of genomic sites in each isolate were mapped by 5 or more reads, indicating high genome coverage (Fig S10). These quality metrics rule out the isolates with missing data or biased read mapping and minimize batch effects. The mpileup files (generated by the SAMtools software) of only these 5977 isolates were further processed for indel and scar analyses. An indel at a genomic site was reported if: (i) the genomic site had depth (number of reads mapped at that genomic site) of 20× or greater; (ii) at least half of the mapped reads (thus, at least 10 reads) supported the indel; (iii) the average mapping quality was at least 10; and (iv) the forward/reverse balance (a measure of strand bias) was ≥0.05 (see ScarTrek pseudocode in Fig. S9, and Figs. S11, S12, and Supplementary Dataset 5 for quality metrics of indels that passed these filters). Forward/reverse balance (as implemented in the CLC workbench) is calculated as: $\min\left(\left(\frac{n_{forward}}{n_{total}}\right),\left(\frac{n_{reverse}}{n_{total}}\right)\right)$, where $n_{forward}$ represents the number of forward reads supporting the indel, $n_{reverse}$ represents the number of reverse reads supporting the indel, and $n_{total}$ represents the total number of reads supporting the indel.

The following downstream quality checks were performed to select high-confidence indels for analyses: (1) All indels were tested for quality based on read-support for the indel using the $\chi^2$ test described in Fang et al.[39]. No "low" quality indels were detected (12.5% of indels were deemed high-quality and 87.5% were moderate quality). Thus, no indels were removed by this step; (2) Indels within a homopolymer region of length 5 bp or more in the reference genome were ignored (59 unique indels); (3) Indels in the repetitive regions such as the MIRU-VNTR regions were ignored (73 unique indels, Table S9); (4) Indels at sites that had biased read-mapping were ignored (indel read depth greater than twice the average genome-wide read depth, 21 such unique indels). Although short reads spanning tens of bases can map incorrectly in repetitive regions of the genome, our dataset consisted of 2 × 100 paired-end reads that improve alignment in repeat-rich regions.

For the 16,693 unique indels remaining after these downstream indel-quality filters, the gene association of each indel was determined using the gene boundaries of the reference *M. tuberculosis* H37Rv (GenBank ID: AL123456.3). For genes that had multiple indels such that at least one indel was a frameshift indel, all the indels were reintroduced simultaneously in the reference gene sequence. The translated gene product from the mutated gene sequence was then compared to the reference protein sequence in the *M. tuberculosis* H37Rv GenPept file corresponding to the GenBank ID AL123456.3. If the combined effect of all indels led to a stop codon in the translated gene product, the gene was ignored for scar analyses. The gene was also ignored if the translated gene product had garbled amino acid sequence from any one of the indel sites up to the end of the gene product. However, if the translated gene product aligned with the reference protein sequence from both ends, and differed only in the region in-between the indels, the set of indels that were introduced in the reference gene sequence were reported as "scar indels" and the gene containing the scar indels was reported as a "scarred gene". Genes that had

only in-frame indels were ignored for scar analyses. The scripts for detecting scars are available at: https://github.com/aditi9783/ScarTrek.

**WGS data analysis pipeline.** The SNPTB bioinformatics pipeline for analyses of WGS data from *M. tuberculosis* samples was used for quality control, read alignment, and single nucleotide polymorphism (SNP) identification with respect to the *M. tuberculosis* reference genome H37Rv (GenBank ID: AL123456.3)[48]. In brief, after inspecting raw read quality using FastQC, Illumina adapters and low-quality ends of reads were removed using Trimmomatic (version 0.36)[49,50]. Reads shorter than 20 bp were dropped and reads were clipped if the average quality score in a window of 4 bp fell below 20. The remaining high-quality reads were mapped to the H37Rv reference genome using Bowtie2 (version 2.2.6)[50,51]. SAMtools (version 1.2) and BCFtools (version 1.2) were then used to identify SNPs in the mapped reads such that the probability of an incorrect SNP call is <1e-20[52,53]. SAMtools was also used to generate mpileup files that contain detailed information about read-alignments in a text format.

**SNPTB and ScarTrek validation.** The SNPTB (for SNP calling) and ScarTrek (for indel and scar calling) code was tested with WGS data (Illumina HiSeq 2500 platform) from three in vitro cultures of laboratory strain of *M. tuberculosis* (H37Rv) grown from the same bacterial stock at different times (data from NCBI SRA BioProject ID accession number PRJNA607763[54]). The three WGS datasets had an average depth of 100X, with >99% of the bacterial genome having ≥20 reads at each genomic position (thus, high genome coverage), and >99% of the high-quality reads mapping to the reference genome (GenBank ID: AL123456.3). The SNPTB pipeline found the three cultures to be genetically identical (ignoring SNPs in the PE/PPE genes) and ScarTrek identified the same set of 17 indels in each of the three cultures (not ignoring any genomic regions) relative to the NCBI reference genome of H37Rv.

**Indel calling using SAMtools.** The VCF (Variant Calling Format) file produced by the SAMtools (version 1.2) step of the SNPTB pipeline was processed to retrieve indels with QUAL score ≥100. These indels were treated as indels predicted by SAMtools.

**Indel calling using GATK HaplotypeCaller.** The GATK (version 4.0.8.1) HaplotypeCaller was used to generate a genomic VCF file, which was then used to generate the VCF file using GenotypeGVCFs function of GATK[55]. The GATK VCF files were then processed to retrieve indels that had QUAL ≥100 and GQ ≥50. The retrieved indels were treated as indels predicted by GATK HaplotypeCaller.

**Reformatting indel calls from SAMtools and GATK HaplotypeCaller.** The indels were reformatted as follows: <ref_base>_<genomic_position>_<type_and_indel_len><indel_string>, where ref_base is the reference genome base at the genomic position where indel was detected, indel "type" is "+" for insertion, and "-" for deletion, and indel_string is the bases that were inserted or deleted. For example: indel with reference string "AAC" and mutated string "A" was noted down as a deletion −2AC. An indel with reference string "AA" and mutated string "AAGCG" was noted down as an insertion +3GCG. ScarTrek detects indels in this format. Indel calling tools were compared by the number of exact matches of predicted indels.

**Evaluation of indel and scar detection by ScarTrek.** From the indels detected in non PE-PPE genes, ScarTrek identified 88 scars where frameshift indels disrupted a reading frame in isolation but maintained it when present together. After manual inspection of the mapped reads for each of the indels in these 88 scars, we confirmed 74 unique scars in 402 isolates that resulted from 155 scar indels, of which 140 were unique (see Fig. S13 for examples of scars that were rejected after manual inspection). We used these 155 indels and 2 additional indels that were confirmed by read inspection as a "gold-standard indel set" to evaluate the performance of ScarTrek against SAMtools and GATK HaplotypeCaller by simulating reads from the *M. tuberculosis* reference genomes in which these 157 indels had been introduced[46,52,55,56]. ScarTrek generally identified indels better than SAMtools and performed similar to GATK HaplotypeCaller for the simulated data (Table S1 and Fig. S1). We further tested the three methods on all the indels in the 402 isolates that had confirmed scars and found that only 2.4% of indels detected by ScarTrek were not found by SAMtools and/or GATK HaplotypeCaller, whereas more than half of indels detected by SAMtools and/or GATK HaplotypeCaller were not supported by the other methods (Fig. S2).

**Generating simulated reads using ART.** The "gold-standard indel set" of 157 indels were introduced in 10 copies of the reference genome. These ten genomes were then used to generate paired-end reads simulated from the Illumina HiSeq 2000 system with the built-in quality score profile that accompanied the NGS read simulator ART (ART_Illumina, Q version 2.5.8)[56]. The read simulation parameters chosen were: read length of 100 with mean fragment size of 300 and a standard

deviation of 30. The default indel error rates were used: first-read insertion rate of 0.00009, second-read insertion rate of 0.00015, first-read deletion rate of 0.00011, and second-read deletion rate of 0.00023. The reads were simulated from the 10-genome dataset at three different settings of fold coverage: 50×, 100×, and 200×. The simulated reads were processed for variant calls using the pipelines described above.

**Computing linguistic complexity.** Linguistic complexity measures the extent to which a sequence contains the non-repetitive combinations of letters from the alphabet[26]. For a sequence of length $n$, its complexity score is defined as $LC = \prod_{i=1}^{n-1} U_i$, where $U_i$ is the ratio of the actual number to the maximum possible number of all combinations of letters in a subsequence of length $i$. The complexity score is between 0 and 1 with low scores indicating the presence of repetitive combinations of letters in the sequence. For DNA sequence, the alphabet is the set of nucleotides. For computing complexity scores for the *M. tuberculosis* H37Rv reference genome, we split the genome into overlapping 21 nucleotide-long windows. Thus complexity score of a given site considers ten positions upstream and downstream of the site in addition to the site itself.

**Computing Shannon's entropy.** Shannon's entropy quantifies the nucleotide diversity of a sequence from the frequencies of letters in the alphabet[27,28]. For a DNA sequence, Shannon's entropy is defined as $H = -\sum_{j=\{A,T,C,G\}} p_j \log_4 p_j$, where $p_j$ is the frequency of nucleotide $j$ in the sequence. Thus, $p_j = n_j/n$, where $n_j$ is the number of times nucleotide $j$ appears in a sequence of length $n$. We split the *M. tuberculosis* H37Rv reference genome into overlapping 21 nucleotide-long windows, as was done for computing linguistic complexity. The Shannon's entropy for a given genomic site thus considered nucleotide frequencies from ten bases upstream to ten bases downstream of the site. By setting the logarithm base to 4, we obtained the entropy values between 0 and 1 with a homopolymer sequence (no nucleotide diversity) having a score of 0 and a sequence with the equal occurrence of all nucleotides getting a score of 1.

**FindingInfo: determining the information content of genomic sequences.** Low-complexity regions of the genome (repetitive sequences and homopolymers) are thought to have low "information content" due to the limitations on how much information about protein structure and function can be encoded by a single nucleotide or by limited combinations of nucleotides. The "FindingInfo" tool computes linguistic complexity and Shannon's entropy for genomic sequences in fasta format, the output of which can be probed for detecting regions of low information content (low $LC$ and $H$ scores) as well as regions of high complexity. FindingInfo is available at: https://github.com/aditi9783/FindingInfo.

**Phylogenetic analysis.** We generated unrooted phylogenetic trees using two approaches: Bayesian and Neighbor-Joining. For the Bayesian method, we followed the general approach outlined in Farhat et al.[57]. In brief, we created a superset of SNPs (ignoring the SNPs in PE-PPE genes and 39 drug-resistance genes[58]) relative to the reference genome H37Rv (GenBank ID: AL123456.3) called in each isolate and generated a multiple sequence alignment by concatenating these SNPs. Phylogenetic trees were then constructed using MrBayes version 3.2.7a[59] with the GTR model and Markov Chain Monte Carlo (MCMC) simulations being run till standard deviation of split frequencies reached <0.05. The MCMC simulations were run for longer (till 300,000 generations) if the convergence diagnostic PSRF for all parameters did not approach 1 (if values were >1.15 or <0.85). This resulted in lower split frequencies as well (average split frequencies from all replicates: 0.02). Phylogenies were also constructed using the Neighbor-Joining method where the mutational distance between two *M. tuberculosis* isolates is defined as the number of SNP differences between them (SNPs in the repetitive PE/PPE genic regions and drug-resistance genes are ignored, as described above). The $n \times n$ distance matrices between $n$ isolates of interest were constructed from their mutational distances and these distance matrices were used to generate neighbor-joining trees using PHYLIP (Phylogeny Inference Package) version 3.696[60]. Phylogenetic trees constructed from MrBayes are shown in all figures unless otherwise noted. Both the Bayesian and Neighbor Joining approaches were in agreement unless otherwise noted.

**Identifying phylogenetically distant isolates.** Pairwise mutational distances between each pair of isolates in our dataset of 5977 isolates were determined. Distance matrix from these pairwise distances was used to generate a Neighbor Joining phylogenetic tree using the PHYLIP program. The output tree file was in the Newick format that was parsed to identify isolates in the tree that are not direct neighbors (separated by at least three intervening clades in the tree) and thus are unlikely to be phylogenetically close. In brief, the Newick format lists immediate evolutionary relatives within the brackets. We selected one isolate from each set of immediate evolutionary relatives such that the selected isolate had largest branch length (and thus the largest evolutionary distance) from the common ancestor of

all the isolates in the clade. In addition, any two independent isolates were separated by at least three nodes in the tree. Thus, if the Newick format of a tree is represented as (A, B, C), D,(((E, F), G), H), where each letter is an isolate and each parenthesis set represents a leaf set in the tree, then isolates from the clades (A, B, C) and (E, F) that had the longest branch length were selected as "phylogenetically distant" isolates. We identified 588 isolates that satisfied these criteria and did not have any scars. We then randomly selected 200 isolates (with replacement) from these 588 isolates for the purpose of visualizing the phylogenetic relationship between isolates that were scarred in a given gene.

**Identifying scar clades**. A scar clade is a set of isolates descended from a "scarred" common ancestor where the scar indels were passed on to each and every isolate in the clade. Since the phylogenetic trees for each gene were constructed with isolates containing the scarred gene and an additional 200 phylogenetically distant isolates that did not have scar indels, the phylogenetic tree (in the Newick format) was read to count the number of scar clades in the presence of non-scarred isolates. Note that a single isolate can constitute one scar clade if it is separated from other scar-isolates in a phylogenetic tree. The number of scar clades seen for all scarred genes was added to determine the incidence of evolutionarily independent scar occurrences. This process was repeated ten times, each with a new random sample of 200 phylogenetically distant isolates, and scar incidences for each replicate are reported. Note that in each replicate, the same set of 200 non-scarred phylogenetically distant isolates were combined with scarred isolates for each gene to generate phylogenetic tree. Since each gene has different number of scarred isolates, the total number of isolates depicted in each gene tree is different, even though they share the same 200 non-scarred phylogenetically distant isolates.

**Taxonomic classification of genomes in isolates with indels in the 16S rRNA (rrs) gene**. Out of 5977 isolates, nine had indels in the *rrs* gene, raising the possibility of mixed infection or contamination. To confirm the presence of *M. tuberculosis* in these isolates, the 1000-fold compressed k-mer signatures ($k = 31$) of these isolates were generated from their raw reads using the "compute" function of package called "sourmash" written for k-mer based taxonomic exploration of genomes and metagenomes[61]. These signatures were then compared with the NCBI GenBank Microbial Genomes search database ($k = 31$, available from sourmash: https://sourmash.readthedocs.io/en/latest/databases.html) containing contigs/scaffolds/genomes of ~100,000 microbial genomes using 'gather' command of sourmash (Table S3).

**Determining lineage of clinical isolates**. The VCF file for each clinical isolate in the dataset was submitted to the program SNP-IT[62] to determine its *M. tuberculosis* lineage (there are seven lineages, 1–7).

**Statistical tests**. The implementations of statistical tests in the SciPy package version 1.0.0 (scipy.stats) were used for computing the Pearson correlation coefficients (function "pearsonr") and for performing the Welch *t*-tests (function "ttest_ind") for comparing population means of distributions that have unequal variances[63].

**Data visualization**. All data analyses plots were generated using the matplotlib package (version 2.1.2)[64]. The phylogenetic trees were visualized using the Newick utilities package (version 1.6)[65]. Although unrooted, the Newick tree visualizations are implicitly rooted at the top node. The genome plot was drawn using Circos (version 0.69)[66]. Joyplots (Fig. S3) were generated using packages ggridges (version 0.5.0), ggplot2 (version 2.2.1), plyr (version 1.8.4), scales (version 0.5.0), and withr (version 2.1.2) on R (version 3.4.3).

**Reporting summary**. Further information on research design is available in the Nature Research Reporting Summary linked to this article.

## Data availability

**Data Collection**. Whole genome sequencing data from 6509 clinical *Mycobacterium tuberculosis* isolates were collected from the following publications: Zhang et al., Nature Genetics[17] (PMID:23995137); Walker et al., Lancet Infectious Diseases[19] (PMID:26116186); Guerra-Assunção et al., Journal of Infectious Diseases[18] (PMID:25336729). All clinical isolates were mapped to the *M. tuberculosis* H37Rv reference genome (GenBank ID: AL123456.3).

**Source Data**. Data underlying all figures have been provided in the accompanying "Source Data" file or the Supplementary Datasets 1–5. Source data are provided with this paper.

## Code availability

All custom code was implemented in Python version 2.7 and was dependent on packages Numpy (version 1.14.1), SciPy (version 1.0.0), and matplotlib (version 2.1.2). The ScarTrek program is available at: https://github.com/aditi9783/ScarTrek[67]. The FindingInfo program is available at: https://github.com/aditi9783/FindingInfo[68].

Bioinformatics pipeline for analysis of WGS data is available at https://github.com/aditi9783/SNPTB[48]. The scripts for data analysis and visualization are available at https://github.com/aditi9783/Scar_manuscript_scripts. All code are publicly available under the MIT or GNU license.

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

## Acknowledgements

Supported by the National Institute of Allergy and Infectious Diseases award U19AI11276. The authors acknowledge the Office of Advanced Research Computing (OARC) at Rutgers, The State University of New Jersey for providing access to the Perceval (under the National Institutes of Health Grant No. S10OD012346) and Amarel clusters and associated research computing resources that have contributed to the results reported here. URL: http://oarc.rutgers.edu.

## Author contributions

A.G. and D.A. designed the study, A.G. performed the data analyses and developed the bioinformatics tools, A.G. and D.A. wrote the manuscript.

## Competing interests
The authors declare no competing interests.
