## [Peer Review File · Nature Communications]

Reviewers' Comments:

Reviewer #4:

Remarks to the Author:

In most manuscripts analyzing NGS datasets of MTB, only SNPs were focused and investigated. Although presence of scars presented in this manuscript are not totally novel and have been observed in previous studies, this piece of work is a systematic analysis of scars and pinpointed the potential evolutionary selection of scars. Therefore, it may trigger a paradigm shift in microbial genomics and the potential impact could not be neglected.

Basically, most questions raised by reviewers have been resolved and the quality of the manuscript is significantly improved. The only point I felt unsatisfied is the rejection of validation by de novo assembly. With appropriate de novo assembly techniques, in my experience most small indels could be resolved. If revision is required, I would like to request at least some random sampling by de novo assembly to cross-check the validity of ScarTrek.

Reviewer #5:

Remarks to the Author:

This manuscript describes two new methods to describe sequence complexity in bacterial genomes (FindingInfo) and to detect pairs of frameshifting and frame restoring indels (ScarTrek). I did not review the original submission, and I was asked to determine whether the concerns raised by a previous reviewer were adequately addressed. The genetic diversity resulting from small indels is often not considered in bacterial population genomic studies, and these methods are important contributions.

The previous reviewer had two major concerns: 1) that the difficulties of accurately calling indels from short read data, particularly in repetitive regions of the genome like the PE-PPE genes, were not adequately discussed and 2) that homoplastic indels in regions of low complexity are evidence of convergent evolution.

The authors have adequately addressed the first concern. The authors have included comparisons of indel calling using their pipeline to two other commonly used variant callers. Additionally, they have included simulations of short read sequencing of H37Rv with indels introduced. Lastly, they have implemented further filtering of indels and removed discussion of PE-PPE genes from the manuscript.

The authors have not adequately addressed the concern that homoplastic indels are evidence of convergent evolution and adaptation. The frequencies of these scar events are quite low (the gene with the highest number of events has scars in <1% of isolates analyzed here). What is the frequency of frameshifting indels observed alone in the data set? Would it be possible to compare the rate of indels in the genes the authors suggest are under adaptation to other regions of the genome with a similar level of sequence complexity? Alternatively, the conclusion that these events are necessarily adaptive could be softened.

I have additional concerns related to the phylogenetic analysis of indels. The authors used a neighbor joining tree with only 200 non-scarred isolates from their data set of >6000 genomes to assess the number of occurrences of these indels. A more sophisticated method of ancestral reconstruction incorporating an evolutionary model into the phylogenetic analysis would provide stronger estimates for the number of occurrences and phylogenetic placement of these indels.

Main text:

Line 52: I assume this is the number of genes in *M. tuberculosis* H37Rv rather than the pan genome of *M. tuberculosis*. This should be clarified, as other strains may encode more/fewer genes.

Line 85-86: The authors included sequencing data with evidence of contamination or mixed infections with a very low threshold for the percentage of reads mapping to H37Rv (20%). Since the threshold for mapped reads that support the presence of the indel is also low (50%), it appears that indels be a result of contamination. This concern was raised by another reviewer as well.

Line 254: Is the abundance of low complexity regions in *M. tuberculosis* unique compared to other species?

Supplement:

Lines 134-138: It is not clear if SNPs in known repetitive regions (PE-PPE) were masked before phylogenetic analysis. Also, the phylogenetic method used here (Neighbor Joining) is not standard for bacterial phylogenetics. A method with an underlying evolutionary model may be more appropriate (RAxML, IQTree, etc.).

Lines 153-162: If only 200 non-scarred isolates were included in the phylogenetic analysis, it cannot be concluded that "scar clades" constitute clades where all descendants encode the indels. Similarly, if isolates with the same indel are separated on the tree by isolates without the indel, it may be the case that the isolates without the indel simply reverted (particularly in low complexity regions where the mutation rate may be high).

Reviewer #6:

Remarks to the Author:

Gupta and colleagues conducted a computational analysis of WGS data from more than 6000 *M. tuberculosis* isolates from patient samples that were sequenced in three prior studies. The analysis focuses on detection of small indels across these strains. The most important finding from this work was an observation of multiple genes that appeared to acquire multiple frame-shifting indels that would together rescue the gene function. The authors hypothesize that this may be an under-appreciated mode of genomic plasticity that could provide selective advantage to *M. tuberculosis* by adaptive mutation under the effect of different selective pressures.

This is a very interesting study. I was asked to review this manuscript principally to weigh in on technical issues relating to the use of a novel computational method (ScarTrek) which was developed specifically to do this indel analysis.

I appreciate the difficulty of indel calling with short read sequencing. For this type of study, in particular, it is important to attempt to remove as many false positive calls as possible to have

confidence in the results, as the main findings are based on a small number of indel calls that form the genic scars. These will naturally be enhanced for false positives, and the accuracy of these calls are key to the main conclusions.

Major Comment 1. Indel calling accuracy.

I concur with the authors, and the data they present, showing that simply running existing tools out of the box (GATK, Samtools) without additional filtering is not an adequate strategy. In this context, ScarTrek, which was custom built for this purpose, could be viewed as one method of providing the additional stringent filtering which is needed.

There is a lot of suggestive evidence in the manuscript to indicate that the indel calling is of generally good quality. For example, the trends with respect to essential genes are encouraging and the frequency differences between genic and intergenic indels.

As shown in table S1, however, false positives remain a concern. The estimated false positive rate for GATK and ScarTrek on simulated data was 20-30% (Samtools was worse).

The Venn diagram in Figure S2 suggests that ScarTrek is likely to be more specific than GATK or Samtools overall, although the trend of ScarTrek making less calls than GATK, which was true in the analysis on real data, was not apparent in the simulated data (Table S1).

The false positive rate for ScarTrek increased with increasing simulated read depth (from 23% at 50x to 27% at 200x, from Table S1), which is somewhat concerning. This can be the hallmark of a method that, when given more data, becomes more confident in its mistakes. Samtools also shows this trend, while GATK does not (GATK FP rate drops slightly with increasing depth).

Given that indel calling is difficult, and that it is hard to control for false positives, my recommendation would be to focus primarily on the 188 indels that are involved in scar formation and are driving the main conclusions of the paper. The important question is whether these calls are enriched for false positives (above the expected 20-30%) or have been more stringently curated and have a lower false positive rate. I would suggest performing some additional quality analyses on the called indels and also specifically focusing on the 188 scar-forming indels.

Specific suggestions:

1. Compute a Venn diagram similar to Figure S2, but showing what fraction of the scar-forming indels fall into each region. It would be reassuring if the vast majority of the scar-forming indels fall into the 36K common overlap indels, and worrying if a large fraction are in the region unique to ScarTrek.
2. Compute a Venn diagram similar to Figure S2, but showing in each segment the false positive rate from the analysis in Table S1.

This looks at how the estimated FP rate changes based on whether 1, 2 or all 3 callers made each call and in which combinations. I am curious, in particular, about the FP rate of the calls made by ScarTrek and GATK but not Samtools. These are a large fraction of the total indels analyzed, and presumably a number of the scar-forming indels are in this category, and if these show a low FP rate in the simulations, this would be reassuring.

3. For each indel call, calculate quality statistics such as:

- a) Number of supporting reads covering the indel position and number of total reads

- b) Fraction of supporting reads covering the indel
- c) The mapping fraction of the sample in which the indel was called (% of *M. tuberculosis* reads)
- d) The study from which the data originated (and sequencing batch within the study if there are multiple)
- e) The mean and median mapping quality of the reads supporting the indel
- f) The mean and median trimmed read length of the reads supporting the indel
- g) You could also include strand bias (as suggested by another reviewer): A 2x2 table of the counts of reads supporting/not-supporting the indel stratified by read direction.

These could be included in the supplementary data set.

I would then suggest looking at the 188 scar-forming indels in comparison to overall distribution of these metrics. It would be reassuring if the scar-forming indels were not outliers on these quality metrics.

4. In the methods ("Evaluation of indel and scar detection by ScarTrek") the authors performed manual review during which they eliminated 13 indel positions (~13%). These can be used as a kind of negative control in the above analyses. It would be reassuring to see that indel calls from these 13 positions (deemed questionable on manual review) are outliers in the above quality statistics, more than the 85 indel positions that were kept.

5. The number of frame-shift vs. non-frame shift indels is a good quality metric, as the authors point out (line 71). The lack of an actual uptick at +/- 3bp is a little worrisome, although there does seem to be some over-representation of indels at +/- 3bp compared to typical distributions of intergenic indels (in humans).

Sometimes this trend is more visible at +/- 6bp and +/- 9bp. It would be nice to include a supplementary figure similar to Figure 1a but with (a) more values on the y axis and (b) removing the intergenic regions or using these regions as a null for comparison.

6. I am not familiar with all of the difficulties of indel calling in patient-derived bacterial samples. One reason I included the mapping fraction of the sample in the list above was out of a concern for the few percent of samples with low *M. tuberculosis* mapping (e.g. below 90%). An alternative approach would be to simply exclude these few samples from the scar analysis, which I assume would have little effect on the results.

7. Use the phylogeny as an additional quality metric.

I thought the phylogenetic analysis was quite interesting. I am puzzled, however, why the authors kept computing new phylogenetic trees among the strains for each gene or each analysis. I would have found it more natural, and easier to follow and interpret, if they had established the most-likely phylogeny once, then mapped all other analyses on to the same phylogeny.

Moreover, consistency of the indels calls with the phylogeny is a very useful, orthogonal quality metric that can be used to evaluate relative false positive rates among different groups of indels. Good indel calls should be generally consistent with the phylogeny, while artifacts often will not be. I don't know offhand the right statistical test for this, but it is similar to the homoplasy analysis (Figure S9) but using a statistical test to determine when indels are mostly consistent with the phylogeny (uncertainty in the phylogeny being the main confounder).

Testing the phylogenetic consistency of each indel (especially the 188 scar-forming indels) would provide an additional quality check.

I'm not suggesting that you should filter out any indels that don't follow the phylogeny, just that most of the indels should follow the phylogeny in expected patterns. Restoring a gene by convergent evolution using the exact same fixing indel would presumably be a rare event.

Major Comment 2: Differential indel rates among classes of genes

Starting around line 143, the authors contrast the rates of unique indels among 3 categories of genes (PE-PPE, "essential" and the rest, labeled as "non-essential"). The authors state that there is a positive correlation with the first category, none with the second, and a "weak" correlation with the third.

I feel like the author's characterizations are perhaps somewhat misleading and I think the differences might be important and interesting.

First, it is not true that there is no correlation with the "essential" genes. Although the correlation is relatively weak in comparison to the other classes, there is still correlation ($r=0.22$, $p=2 \times 10^{-6}$). The "other" genes (also referred to here as "non-essential") have an intermediate correlation between the essential and PE-PPE genes. Moreover, although it is a little hard to see, when I look at the scatter plot in Figure 3, it looks like the "other" genes may be a mixture between two groups of genes, some of which are more like the essential genes and some of which are more like the PE-PPE genes.

This all seems plausible to me. As the authors note, the definition of "essential" used here is from an in-vitro assay and not necessarily representative of the life cycle of wildtype *M. tuberculosis*, although one would hope that this set is highly enriched for essential genes in all contexts. The "other" genes may well be a mixture of genes that are both essential and non-essential. I note that loss-of-function intolerance, for example as measured by the pLI score, has emerged as an important tool in human genetics. Perhaps the indel rates shown here are similar evidence of which *M. tuberculosis* genes are more highly constrained.

Minor Comments:

1. While I find this a very interesting study, in a number of places I would prefer the authors to not overstate the evidence for selection. As one example, in the Abstract, I might suggest "Nineteen scars showed evidence of consistent with (or suggestive of) convergent evolution. Our results suggest that sequentially occurring indels are may be an evolutionary mechanism for short-term gene silencing and adaptation in *M. tuberculosis*." These kinds of statements appear through the manuscript.

2. Phylogeny analysis. As mentioned, I would have preferred that all of the analyses were carried out against a fixed phylogeny (downsampling for display purposes, but always in the same way). For example in Figures S6 to S9, it is confusing to try to compare the analyses in different panels. Much of the time one can map the phylogenies to each other by eye, but sometimes the phylogenies in some subpanels look quite different, presumably due to the sampling.

Moreover, I would have liked to see more analysis showing the relationship of the two (or more) indels forming a scar. The authors indicate that when a pair of indels form a scar event, one (or sometimes both?) of these indels is sometimes seen by itself in other samples, as would be expected. One might expect to observe a number of cases where the first ("wounding") indel occurs near the root of the tree and then second ("fixing") indel occurs one or more times closer to the leaves of the tree. The branch lengths in the phylogeny would provide information about the timing of these indels and perhaps additional evidence about whether there is any selective pressure to "correct" a frame shift in

a particular gene.

3. Figure 6. I like Figure 6 quite a lot, and the phylogenetic analysis in general.

However, I have a number of suggested improvements:

a) Rather than identifying the scars by number (1-10) on the Y axis of panel (b), I would suggest giving them letters A-J.

b) I find it hard to match up the events on panel (a) with panel (b) by matching the colors (where is the yellow event that is visible on panel (a)?). I would suggest that in addition to using color, label the events on panel (a) with the letters (A-J) of the scar on panel (b) near the event in order to make the correspondence unambiguous. You could also refer to these labels in the text (instead of referring to the scars by color), which would make the work more accessible to people with limited color perception.

c) I want to know the frequency of the events more clearly. It is clear that the gray event is of much higher frequency and has expanded in a particular clade. Perhaps you could write the number of strains where the scar was detected to the right of each scar on panel (b). Even for the less frequent events, I would like to know if they are singletons, doubletons, etc.

d) It is not obvious when two indel sites are shared or are simply close by. Perhaps this could be indicated somehow visually (perhaps a vertical dashed line?)

e) Assuming the last two indels are the same, the purple and red events (numbers 2 and 9) look likely to be derived from the gray event (number 5). I couldn't tell if this matched the phylogeny.

f) If the last two indels in scars number 2, 5 and 9 are the same, then since the gray event restores frame, the middle segment of the purple event must also be in frame. This suggests that the purple event is actually two small scars, not one long one. And the same for the red event, although the situation here might be more complicated. Two short scars also increase the evidence that the scars in this gene may be under selective pressure to be short.

4. Regarding Figure S2, comparing indel calls between tools is notoriously difficult, due to different callers choosing slightly different representations for the indel. The authors didn't mention how they handled this. Did you use a normalization tool? Did you use a less restrictive test than an exact allelic match?

5. In the methods section "Evaluation of indel and scar detection by ScarTrek". Near the end it says "5.2% of indels detected by ScarTrek were not found by SAMtools and GATK HaplotypeCaller". You should say "SAMtools and/or GATK", as you do in the main text, and be consistent with the language in the rest of that paragraph.

6. In this same section: At the end, the text claims ScarTrek has 86.7% accuracy. I was unable to discern how you derived this number. I would also avoid using the word "accuracy" based on comparisons between callers. If this is derived somehow from the number of overlapping calls, I would just explain how you derived it.

7. Figure 5a does not appear to be referenced from the main text.

8. Line 97: Says "frame-shifting indels were uniformly distributed along the gene length for the PE-PPE and 98 non-essential genes (Fig. 1c)". However according to the figure description, Figure 1c shows all indels, not just frame-shifting indels. So either the legend is wrong or Fig 1c does not speak directly to this statement.

9. Figure 3: Could use a legend to indicate the meaning of the colors rather than just putting this in the figure description.

10. I really liked supplementary Figure S3.

REVIEWER #4:

Critique: In most manuscripts analyzing NGS datasets of MTB, only SNPs were focused and investigated. Although presence of scars presented in this manuscript are not totally novel and have been observed in previous studies, this piece of work is a systematic analysis of scars and pinpointed the potential evolutionary selection of scars. Therefore, it may trigger a paradigm shift in microbial genomics and the potential impact could not be neglected. Basically, most questions raised by reviewers have been resolved and the quality of the manuscript is significantly improved.

Response: We thank the reviewer for the positive review.

Critique: The only point I felt unsatisfied is the rejection of validation by de novo assembly. With appropriate de novo assembly techniques, in my experience most small indels could be resolved. If revision is required, I would like to request at least some random sampling by de novo assembly to cross-check the validity of ScarTrek.

Response: We focused on SAMtools and GATK for comparison and validation of ScarTrek because these tools call indels from reads aligned to the reference genome, just like ScarTrek, allowing for a direct comparison of performance. GATK realigns reads in the vicinity of a variant to optimize indel calling using a “hybrid” approach that incorporates both read alignment and local de novo assembly, an approach that is common in several new indel callers. The high concordance between ScarTrek and GATK in simulated data (see figures S1 and S2) provides confidence of ScarTrek’s ability to detect indels compared to other methods that include local de novo assembly.

Furthermore, our local alignment approach is justified by a body of literature which shows that de novo assembly methods are more likely to detect large indels but local alignment is more likely to detect short indels of length <50 bp that were the main subject of our study. We now discuss indel calling by read-alignment versus de novo assembly approaches in the Discussion section of the manuscript. We respectfully suggest that a further broader study comparing indel calling by de novo approaches is out of scope for our manuscript.

Finally, please note that reviewer #5 below has concluded that in regards to prior critiques about “1) the difficulties of accurately calling indels from short read data, particularly in repetitive regions of the genome like the PE-PPE genes...”, “the authors have adequately addressed [this] concern...”.

REVIEWER #5:

Critique: This manuscript describes two new methods to describe sequence complexity in bacterial genomes (FindingInfo) and to detect pairs of frameshifting and frame restoring indels (ScarTrek). I did not review the original submission, and I was asked to determine whether the concerns raised by a previous reviewer were adequately addressed. The genetic diversity

resulting from small indels is often not considered in bacterial population genomic studies, and these methods are important contributions.

Response: We thank the reviewer for recognizing the important contributions of our work.

Critique: The previous reviewer had two major concerns: 1) that the difficulties of accurately calling indels from short read data, particularly in repetitive regions of the genome like the PE-PPE genes, were not adequately discussed and 2) that homoplastic indels in regions of low complexity are evidence of convergent evolution.

The authors have adequately addressed the first concern. The authors have included comparisons of indel calling using their pipeline to two other commonly used variant callers. Additionally, they have included simulations of short read sequencing of H37Rv with indels introduced. Lastly, they have implemented further filtering of indels and removed discussion of PE-PPE genes from the manuscript.

The authors have not adequately addressed the concern that homoplastic indels are evidence of convergent evolution and adaptation. i) The frequencies of these scar events are quite low (the gene with the highest number of events has scars in <1% of isolates analyzed here).

Response: We agree that low sequence complexity can lead to higher than expected indel occurrence that is not due to selection pressures. However, scars are multi-indel events, and observing same scars in evolutionarily distinct isolates raises the possibility of homoplasmy in scar indels. To further investigate the relationship between sequence complexity and homoplasmy in indels, we analyzed the sequence composition and phylogenetic distribution of indels in very low and very high sequence complexity regions and found some indels in high sequence complexity regions to be homoplastic, even though “homoplasmy” was three times more common in low complexity regions (Suppl. Fig. S8). Thus, while not all homoplastic indels suggest convergent evolution, observing homoplasmy in multi-indel events such as scars in non-repetitive genes does raise that possibility. We have further clarified this in the revision.

We agree with the reviewer that the frequency of scar events in any individual gene appear to be low. However, the overall frequency of scar events in any individual *M. tuberculosis* genome are substantially higher, as we detected a 4.47% population-level incidence of unique scar events. Unlike mutations associated with drug resistance, which map to a very small numbers of genes and pathways directly related to drug entry or drug mechanism of action, it is perhaps not surprising that genetic events postulated to cause variances in the pathogenesis and immune responses in clinical settings might map to many more genes and biological pathways given the multitude of evolutionary paths that can provide similar adaptive benefits. Thus, it is the population-level incidence of unique scars that is probably the more relevant number when discussing the frequency and significance of scar formation.

Further, as we note in the discussion, ScarTrek likely under-reports scars because it looks at a single genomic snapshot. For example, an insertion of length 1 followed by an insertion of 2

bases adjacent to the 1st indel would appear as an in-frame insertion of 3 bases. In fact, such scar events were detected *in vitro* by Safi *et al.* (*Proc Natl Acad Sci U S A* 2019, reference 15 in the main text). Indeed, in absence of longitudinal data from same patients, it is impossible to quantify the true frequency of scars.

Critique: ii) What is the frequency of frameshifting indels observed alone in the data set?

Response: This is an important question and we apologize for omitting it in our original submission. Analyzing our data, we found that 81% of all indels were frame-shift indels. This is now reported in the Results section “Indels cause substantial genomic plasticity in *M. tuberculosis*”.

Critique: iii) Would it be possible to compare the rate of indels in the genes the authors suggest are under adaptation to other regions of the genome with a similar level of sequence complexity? Alternatively, the conclusion that these events are necessarily adaptive could be softened.

Response: In answer to the reviewer’s question, we report in the Results section that the number of unique indels in scarred genes is similar to non-scarred genes that had comparable low-complexity gene fraction (Figures 5c-d). Thus, the presence of scars in some genes does not in itself predict the frequency of indel formation. Instead, as we discuss and show in Figure 2a-c, it is the sequence composition of a very local region distinguishes indel sites from non-indel sites in the gene.

The important point here, is that we do not believe that the presence of indels can be used to identify genes that are under the type of reversible adaptive pressure that should lead to the reversible indels indicated by scar formation. Instead, we state in the revised manuscript that “while overall gene complexity influences indel occurrence, it may not determine whether scars form in certain genes. This suggests that the pool of genes that can contribute to adaptation through wounding and scarring is potentially large, and the biological effect of a frame-shift wounding indel is likely to be the most important predictor of whether a particular gene acquires a frame-shift scar.”

To reiterate, our results suggest that it is the presence of multi-indel events like scars and not isolated indels that indicate which genes are under the type of adaptive pressure that requires reversibility of gene function. Unlike a simple indel, the analysis of reversible mutations in the *glpK* gene by Safi *et al.* (PNAS, 2019) suggests that scars might indicate two events, a “wounding” indel that disrupts an open reading frame, essentially inactivating a gene, and a second “fixing” indel that restores the frame and the function of the gene. One purpose of our manuscript was to analyze the entire *M. tuberculosis* genome for genes that might be subject to this type of adaptive response. Further, identifying genes or gene clusters with a higher number of scars might indicate loci where this type of plasticity was particularly important. It is for this reasons that we believe our finding of multiple scars in the ESX-1 gene cluster (Fig. 4 and related text) constitutes an important finding regarding adaptive fitness in *M. tuberculosis*.

However, despite the conclusions that we believe are derived from our results, we understand the reviewers request for caution in our interpretations. While we do see convergent evolution of *some* scars and scar-indels (and thus likely under adaptation), we find more scars that appear in close evolutionary relatives only, or appear only once in the dataset. We have now made this clear in the abstract and the results sections as follows: “Based on phylogenetic analysis, eight scars evolved in multiple clinical isolates likely by convergent evolution, although scars descending from an evolutionary ancestor were more common”. We hope this new more cautious wording will be acceptable to the reviewer.

Critique: I have additional concerns related to the phylogenetic analysis of indels. The authors used a neighbor joining tree with only 200 non-scarred isolates from their data set of >6000 genomes to assess the number of occurrences of these indels. A more sophisticated method of ancestral reconstruction incorporating an evolutionary model into the phylogenetic analysis would provide stronger estimates for the number of occurrences and phylogenetic placement of these indels.

Response: The phylogenetic tree drawing program was unable to render a tree with a large number of isolates while also depicting different scarred isolates in different colors. Thus, we used the following method for detecting occurrences and phylogenetic placements of scar indels while considering as many isolates as possible from our dataset:

Step 1: For each scarred gene, we constructed a phylogenetic tree of the isolates that had the scarred gene along with 200 randomly selected phylogenetically distant isolates in our entire dataset of ~6000 isolates (see “identifying phylogenetically distant isolates” in Methods). The goal was to see if the isolates that have scars in a given gene are phylogenetically close against a background of 200 isolates that are not scarred. The figure 6a shows one such tree for the gene *espl*.

Step 2: Such a tree was created for *each* of the 49 scarred genes using the same set of 200 randomly selected phylogenetically distant isolates.

Step 3: The total number of scar clades (independent scar occurrences) in these 49 trees (see section “identifying scar clades” in the online methods) were reported.

This process (steps 1 to 3) was repeated 10 times with a new random sample of 200 phylogenetically distant isolates in each tree, and the total number of scar clades from these 10 iterations was reported in the main manuscript. We realize that we only referenced this approach as “online methods” in the main text, due to space constraints. However, we have now included the following description in the main text: “online methods, in brief: for each scarred gene, a phylogenetic tree was constructed from isolates that had a scar in that gene along with 200 phylogenetically distant isolates, and total number of scar clades from all the trees are reported”.

Critique: Main text: Line 52: I assume this is the number of genes in *M. tuberculosis* H37Rv rather than the pan genome of *M. tuberculosis*. This should be clarified, as other strains may encode more/fewer genes.

Response: We apologize for this omission. We have included the strain clarification as suggested by the reviewer.

Critique: Main text: Line 85-86: The authors included sequencing data with evidence of contamination or mixed infections with a very low threshold for the percentage of reads mapping to H37Rv (20%). Since the threshold for mapped reads that support the presence of the indel is also low (50%), it appears that indels be a result of contamination. This concern was raised by another reviewer as well.

Response: Tuberculosis is often a chronic disease that can be accompanied by extensive lung destruction. It is not uncommon for other bacteria including slow growing bacteria and non-tuberculosis mycobacteria species that could be co-cultured with *M. tuberculosis* to colonize or secondarily infect this diseased lung at the same time as the *Mycobacterium tuberculosis* infection is taking place. Thus, it is also not uncommon for a large number of reads to go unmapped against the *M. tuberculosis* genome in cultures of this type. Thus, we did not want to restrict ourselves to only those isolates that have >90% of reads mapped to the reference. In response to the reviewer's concerns, we have now increased the read mapping % from 20% to 50%. It should be noted that the read mapped % is not as strong a filter as the read depth (number of reads mapped at a position): a low yield sequencing run that generated only 100,000 reads will be insufficient for the Mtb genome that spans ~ 4 million bases, even with 100% mapping rate. Thus, we have included a second filter that required an average read depth of 20 or more at every genomic position to be considered for indel calling. Further, an indel was only called if at least 50% of the mapped reads contained the same indel (thus, each indel call is supported by at least 10 reads). Supplementary figures S9 and S10 show that majority of samples had >90% read mapping rate and 90% read support for indel calls, indicating the high quality of the dataset used.

These stringent rules decreased our dataset to 5,978 isolates and also removed 5 of the 14 isolates that had indels in the *rrs* gene. The possibility of contamination was investigated in detail by metagenomics analysis for the remaining 9 isolates that had indels in the *rrs* gene (see section called "Taxonomic classification of genomes in isolates with indels in the 16S rRNA (*rrs*) gene" in the online methods, and Table S3). All of these isolates mapped to *M. tuberculosis* as the dominant and only hit (Table S3).

To summarize: i). We increased the read mapping % threshold to 50%. This resulted in removal of only 1% of our dataset (68 isolates). ii) We added a new filter that removed isolates that had average genome-wide depth of less than 20 (i.e., on average, each position in the genome was mapped by 20 reads or more). This new filter removed 337 isolates (5.3% of our dataset). The final dataset now has 5,978 isolates. iii) This reduced dataset had only 9 isolates that had indel

in *rrs* gene, but all of these isolates had *M. tuberculosis* as the major hit in metagenomics analysis.

Critique: Main text: Line 254: Is the abundance of low complexity regions in *M. tuberculosis* unique compared to other species?

Response: Our focus in this study was to analyze sequence complexity of *M. tuberculosis* as the first step towards investigating the role on indels in bacterial adaptation. The next step would be to carry out similar analysis for other bacteria in the *M. tuberculosis* complex, however, this was not the immediate focus of this study and remains a future study goal.

Critique Supplement: Lines 134-138: It is not clear if SNPs in known repetitive regions (PE-PPE) were masked before phylogenetic analysis. Also, the phylogenetic method used here (Neighbor Joining) is not standard for bacterial phylogenetics. A method with an underlying evolutionary model may be more appropriate (RAxML, IQTree, etc.).

Response: i) We thank the reviewer for pointing this out. We had not masked the SNPs in the repetitive regions (PE PPE genes) in our original analysis. We have fixed this issue now and all the phylogenetic analysis has been redone with the new SNP counts ignoring the PE PPE genes. ii) Neighbor joining method of tree construction is commonly used in *M. tuberculosis* phylogenetic studies such as polyphasic genotyping (Weniger et al, 2010, Nucleic Acids Research), epidemiological studies (Wirth et al, 2008, Plos Pathogens), and investigations into the genetic diversity of the Mtb bacterial complex (Coscolla et al, 2013, Emerging Infectious Diseases). Due to this precedence, we used NJ method in this work as well.

Critique: Supplement: Lines 153-162: If only 200 non-scarred isolates were included in the phylogenetic analysis, it cannot be concluded that “scar clades” constitute clades where all descendants encode the indels. Similarly, if isolates with the same indel are separated on the tree by isolates without the indel, it may be the case that the isolates without the indel simply reverted (particularly in low complexity regions where the mutation rate may be high).

Response: As mentioned in response to an earlier comment, the phylogenetic analysis of the 49 scarred genes was carried out 10 times, each time with a new random sample of 200 non-scarred and phylogenetically distant isolates to determine number of scar occurrences and their phylogenetic placements. As for the reviewer’s second point, scarred clades that we identified as distinct events were separated from another clade with the same scar by multiple not-scarred clades, and only shared a distant common ancestor with the other scarred clade, making a single reversion event an unlikely cause for the development of two different scarred clades.

REVIEWER #6:

Critique: This is a very interesting study. I was asked to review this manuscript principally to weigh in on technical issues relating to the use of a novel computational method (ScarTrek) which was developed specifically to do this indel analysis.

Response: We thank the reviewer for the encouraging comment, as well as the useful advice provided below.

Critique:

Major Comment 1. Indel calling accuracy.

I concur with the authors, and the data they present, showing that simply running existing tools out of the box (GATK, Samtools) without additional filtering is not an adequate strategy. In this context, ScarTrek, which was custom built for this purpose, could be viewed as one method of providing the additional stringent filtering which is needed. There is a lot of suggestive evidence in the manuscript to indicate that the indel calling is of generally good quality. For example, the trends with respect to essential genes are encouraging and the frequency differences between genic and intergenic indels. As shown in table S1, however, false positives remain a concern. The estimated false positive rate for GATK and ScarTrek on simulated data was 20-30% (Samtools was worse). The Venn diagram in Figure S2 suggests that ScarTrek is likely to be more specific than GATK or Samtools overall, although the trend of ScarTrek making less calls than GATK, which was true in the analysis on real data, was not apparent in the simulated data (Table S1). The false positive rate for ScarTrek increased with increasing simulated read depth (from 23% at 50x to 27% at 200x, from Table S1), which is somewhat concerning. This can be the hallmark of a method that, when given more data, becomes more confident in its mistakes. Samtools also shows this trend, while GATK does not (GATK FP rate drops slightly with increasing depth). Given that indel calling is difficult, and that it is hard to control for false positives, my recommendation would be to focus primarily on the 188 indels that are involved in scar formation and are driving the main conclusions of the paper. The important question is whether these calls are enriched for false positives (above the expected 20-30%) or have been more stringently curated and have a lower false positive rate. I would suggest performing some additional quality analyses on the called indels and also specifically focusing on the 188 scar-forming indels.

Specific suggestions:

1. Compute a Venn diagram similar to Figure S2, but showing what fraction of the scar-forming indels fall into each region. It would be reassuring if the vast majority of the scar-forming indels fall into the 36K common overlap indels, and worrying if a large fraction are in the region unique to ScarTrek.

Response: We have filtered more of our dataset based on the following two rules: (i) increased read mapping % to 50%; and (ii) incorporated a new filter that ignores isolates that have average genome-wide coverage < 20. These rules removed ~6% of isolates that were originally included in the study. In the new dataset, we detect 75 scars in 49 genes. Some of the scars in our earlier submission had noted all indels found in the scarred gene in an isolate (including the in-frame indels). We have now removed all in-frame indels from scar definitions in Table S6

(since scars arise by frame-shift indels only). Thus, we now have 157 scar indels of which 142 indels are unique (since some scars share indels).

In response to the reviewer's comment, we have now included Figure S2b that depicts the distribution of 157 scar indels in the 414 isolates with scarred genes, as predicted by the three tools (ScarTrek, GATK, and Samtools). Since any indel can become a scar indel if a second (or a set of other indels) fix the reading frame disrupted by the first indel, we do not expect scar indels to have any special properties that distinguishes them from non-scar indels. As expected, we see similar distribution of indel calling by the three tools as in the Figure S2a. The vast majority of scar indels are indeed called by all three methods, as expected by the reviewer.

Critique: Compute a Venn diagram similar to Figure S2, but showing in each segment the false positive rate from the analysis in Table S1. This looks at how the estimated FP rate changes based on whether 1, 2 or all 3 callers made each call and in which combinations. I am curious, in particular, about the FP rate of the calls made by ScarTrek and GATK but not Samtools. These are a large fraction of the total indels analyzed, and presumably a number of the scar-forming indels are in this category, and if these show a low FP rate in the simulations, this would be reassuring.

Response: We apologize for using the term “false positive rate” in the earlier version of the manuscript, which is defined as $FP/(FP+TN)$. It is rather difficult to compute the “FP rate” for indel calling because of the ambiguity in defining a true negative for indel calling (do all positions in the genome that do not have an indel in any of the isolates count as a true negative?). Thus, we have replaced all mentions of FP rate with “number of FP”.

At the reviewer's request, and to provide a more detailed analysis of the indel prediction performance on simulation data in Table S1, we now include Figure S1 that shows the indel calls made by ScarTrek and GATK in isolation and in combination. At low coverages, ScarTrek detects more indels correctly, while GATK HaplotypeCaller minimizes incorrect calls with increasing coverage.

Critique: For each indel call, calculate quality statistics such as:

- a) Number of supporting reads covering the indel position and number of total reads
- b) Fraction of supporting reads covering the indel
- c) The mapping fraction of the sample in which the indel was called (% of *M. tuberculosis* reads)
- d) The study from which the data originated (and sequencing batch within the study if there are multiple)
- e) The mean and median mapping quality of the reads supporting the indel
- f) The mean and median trimmed read length of the reads supporting the indel
- g) You could also include strand bias (as suggested by another reviewer): A 2x2 table of the counts of reads supporting/not-supporting the indel stratified by read direction.

These could be included in the supplementary data set.

I would then suggest looking at the 188 scar-forming indels in comparison to overall distribution of these metrics. It would be reassuring if the scar-forming indels were not outliers on these quality metrics.

Response: Although there are 16,693 unique indels detected in our dataset of 5,978 isolates, it should be noted that each of these unique indels occur in at least one, and often in many, of the ~6,000 isolates. Thus, after counting all indel occurrences predicted by ScarTrek, there are >500,000 indels in our dataset, each with the statistics of read-depth, fraction of reads supporting each indel, etc. Instead of providing raw data of this magnitude, we have provided figures that help visualize the data quality, which appears to be the reviewer's objective. Thus, to follow the reviewer's request and allow a quick assessment of quality statistics of this indel data, we included the following visualizations:

- i) Total number of reads at an indel position, number of reads supporting indel at the indel position, and fractions of reads supporting indel at a indel position (Figure S10a-c, this figure was included in our original submission but has been updated to reflect the new dataset).
- ii) The read mapping % of isolates (i.e. % of reads mapped to *M. tuberculosis*). This figure, Fig. S9(a), was included in the original submission but has been updated for the new dataset. Panel b of Fig. S9 shows the average per-site coverage in each isolate, which is the new filter we included in this revision to ensure our dataset has high-quality data with high genome wide coverage.
- iii) The mean mapping quality of reads at each indel position in each isolate is plotted in Fig. S10d.
- iv) The strand bias at indel positions in all isolates is shown in Fig. S10e. Although we used forward/reverse balance (as implemented in CLC workbench) to formulate a rule to ignore indel calls with strand bias, in the figure S10e, we plot a calculation of strand bias that varies between [0,1] that is easier to interpret. This strand bias is defined as " $sb = \text{abs}(\text{numf} - \text{numr}) / (\text{numf} + \text{numr})$ ", where *numf* and *numr* are the number of forward and reverse reads supporting an indel. Thus, if *numf* = *numr* (i.e., equal support on forward and reverse reads), *sb* = 0. If all of the read support comes from either the forward reads (*numr* = 0) or the reverse reads (*numf* = 0), then *sb* = 1. Thus *sb* ranges from [0,1]. A value of *sb* closer to 1 denotes strand bias" (reproduced from legend of Fig. S10e).

We would like to point out that the indels were called from mpileup output of SAMtools, that ignores anomalous read pairs (i.e. forward and reverse reads that do not map appropriately), and requires a minimum base quality of 13 for each mapped base. We also ignored all indel calls with average mapping quality ≤ 10 at the indel position. In addition, we had trimmed reads via Trimmomatic that dropped low quality ends/regions of reads. Further, all reads less than 20 bp were also dropped by Trimmomatic. Together with assessment of mapping quality and strand bias, we hope the reviewer would agree that we have provided an exhaustive view of the data quality in general, and of read mapping at indel positions in particular. We do not see read length as a quality metric for reads >20 nt (reads smaller than that are already dropped). Thus, lengths of reads at indel positions are not explored.

The three studies from which the samples originated are cited in online methods and the data are available publicly. Further, the number of samples from each study that had scarred genes are mentioned in Table S6.

Critique: In the methods ("Evaluation of indel and scar detection by ScarTrek") the authors performed manual review during which they eliminated 13 indel positions (~13%). These can be used as a kind of negative control in the above analyses. It would be reassuring to see that indel calls from these 13 positions (deemed questionable on manual review) are outliers in the above quality statistics, more than the 85 indel positions that were kept.

Response: We respectfully disagree with the reviewer that these 13 scars (thus multiple indels in each of these scars) constitute a negative control. Visual inspection of reads mapped to the indels of the scars that were eliminated revealed irregularities, mainly: scar indels being in different reads and not appearing together in any read, and some nearby scar indels were combined as one big indel in some reads but not others. We have now compiled a few representative examples of such scars in Supplementary Figure S12 (see panel a for an example of an accepted scar, and panels b and c for examples of rejected scars). A couple of scars were further ignored because the isolates in which they were present did not meet the new read mapping % threshold and other quality thresholds. However, the main reason why we do not agree that these 13 scars would constitute a negative control is because any indel can become a scar indel by a frame-restoring second indel. Thus, all indels that are not scar indels are potential negative controls. For this reason, we do not expect scar indels to have any properties (in terms of read/mapping quality or sequence composition or relative position of indel within a gene) that would distinguish them from indels that are so far non-scarred.

Critique: The number of frame-shift vs. non-frame shift indels is a good quality metric, as the authors point out (line 71). The lack of an actual uptick at +/- 3bp is a little worrisome, although there does seem to be some over-representation of indels at +/- 3bp compared to typical distributions of intergenic indels (in humans).

Sometimes this trend is more visible at +/- 6bp and +/- 9bp. It would be nice to include a supplementary figure similar to Figure 1a but with (a) more values on the y axis and (b) removing the intergenic regions or using these regions as a null for comparison.

Response: We did not expect a strong uptick of in-frame indels because insertion or deletion of a single or more amino acids (even while keeping the reading frame intact) can still alter the protein. We have now included the following statement: "In-frame indels were not significantly abundant likely because, unlike synonymous base substitutions, in-frame indels still alter translated protein sequences (add or remove amino acids), and thus may have an appreciable fitness cost."

In response to the reviewer's request, we have now included frequencies of all indel lengths (Table S10), that shows some over-representation of indels at +/- 6bp and +/- 9bp, like seen at +/- 3bp in the Fig. 1.

Critique: I am not familiar with all of the difficulties of indel calling in patient-derived bacterial samples. One reason I included the mapping fraction of the sample in the list above was out of a concern for the few percent of samples with low *M. tuberculosis* mapping (e.g. below 90%). An alternative approach would be to simply exclude these few samples from the scar analysis, which I assume would have little effects on the results.

Response: Tuberculosis is often a chronic disease that can be accompanied by extensive lung destruction. It is not uncommon for other bacteria including slow growing bacteria and non-tuberculosis mycobacteria species that could be co-cultured with *M. tuberculosis* to colonize or secondarily infect this diseased lung at the same time as the *Mycobacterium tuberculosis* infection is taking place. Thus, it is also not uncommon for a large number of reads to go unmapped against the *M. tuberculosis* genome in cultures of this type. Thus, we did not want to restrict ourselves to only those isolates that have >90% of reads mapped to the reference. We have now increased the read mapping % to 50%. It should be noted that the read mapped % is not as strong a filter as the read depth (number of reads mapped at a position): a low yield sequencing run that generated only 100,000 reads will be insufficient for the Mtb genome that spans ~ 4 million bases, even with 100% mapping rate. Thus, we have included a second filter that required a read depth of 20 or more to be considered for indel calling. Further, an indel was only called if at least 50% of the mapped reads contained the same indel (thus, each indel call is supported by at least 10 reads). Supplementary figures S9 and S10 show that majority of samples had >90% read mapping rate and 90% read support for indel calls, indicating the high quality of the dataset used. To summarize:

- i). We increased the read mapping % threshold to 50%. This resulted in removal of only 1% of our dataset (68 isolates).
- ii) We added a new filter that removed isolates that had average genome-wide depth of less than 20 (i.e., on average, each position in the genome was mapped by 20 reads or more). This new filter removed 337 isolates (5.3% of our dataset).

The final dataset now has 5,978 isolates.

Critique: Use the phylogeny as an additional quality metric. I thought the phylogenetic analysis was quite interesting. I am puzzled, however, why the authors kept computing new phylogenetic trees among the strains for each gene or each analysis. I would have found it more natural, and easier to follow and interpret, if they had established the most-likely phylogeny once, then mapped all other analyses on to the same phylogeny.

Response: We computed the phylogeny for each scarred gene along with 200 randomly selected phylogenetically distant isolates in our entire dataset of ~6000 isolates (see “identifying phylogenetically distant isolates” in supplementary methods), and then repeated this 10 times as a preferred method compared to selecting the most likely phylogeny one single time. The total number of scar clades as well as their average from these 10 iterations was reported in the

main manuscript. This was done to ensure that the tree was not biased by a single sampling of the 200 randomly selected control isolate sequences.

Critique: Moreover, consistency of the indels calls with the phylogeny is a very useful, orthogonal quality metric that can be used to evaluate relative false positive rates among different groups of indels. Good indel calls should be generally consistent with the phylogeny, while artifacts often will not be. I don't know offhand the right statistical test for this, but it is similar to the homoplasy analysis (Figure S9) but using a statistical test to determine when indels are mostly consistent with the phylogeny (uncertainty in the phylogeny being the main confounder). Testing the phylogenetic consistency of each indel (especially the 188 scar-forming indels) would provide an additional quality check.

I'm not suggesting that you should filter out any indels that don't follow the phylogeny, just that most of the indels should follow the phylogeny in expected patterns. Restoring a gene by convergent evolution using the exact same fixing indel would presumably be a rare event.

Response: As can be seen in Fig. 6 almost all of the scars were phylogenetically consistent. We agree that the absence of phylogenetic consistency in all of the scars would have raised concerns about quality and the presence of this consistency is reassuring. However, some scars (8 in our total study set) did show some homoplasy. We suggest that the occurrence of homoplasy in these scars is evidence of independent evolutionary selection, and not low quality. We wished to use phylogeny to understand evolution of scars and scar indels. Thus, homoplasy cannot be used as a quality metric.

Critique: Major Comment 2: Differential indel rates among classes of genes

Starting around line 143, the authors contrast the rates of unique indels among 3 categories of genes (PE-PPE, "essential" and the rest, labeled as "non-essential"). The authors state that there is a positive correlation with the first category, none with the second, and a "weak" correlation with the third.

I feel like the author's characterizations are perhaps somewhat misleading and I think the differences might be important and interesting. First, it is not true that there is no correlation with the "essential" genes. Although the correlation is relatively weak in comparison to the other classes, there is still correlation ($r=0.22$, $p=2 \times 10^{-6}$). The "other" genes (also referred to here as "non-essential") have an intermediate correlation between the essential and PE-PPE genes. Moreover, although it is a little hard to see, when I look at the scatter plot in Figure 3, it looks like the "other" genes may be a mixture between two groups of genes, some of which are more like the essential genes and some of which are more like the PE-PPE genes. This all seems plausible to me. As the authors note, the definition of "essential" used here is from an in-vitro assay and not necessarily representative of the life cycle of wildtype *M. tuberculosis*, although one would hope that this set is highly enriched for essential genes in all contexts. The "other" genes may well be a mixture of genes that are both essential and non-essential. I note that loss-of-function intolerance, for example as measured by the pLI score, has emerged as an

important tool in human genetics. Perhaps the indel rates shown here are similar evidence of which *M. tuberculosis* genes are more highly constrained.

Response: We apologize for our phrasing of this result and agree with the reviewer that the rate of unique indels is strongly correlated with gene length for the PE-PPE genes, moderately correlated with the non-essential and non-PPE genes, and weakly correlated with the essential genes. We have now edited the text (as well as Figure 3 legend) to reflect this: "Moreover, the number of unique indels was strongly correlated with gene length for PE-PPE genes (Fig. 3, Pearson correlation coefficient and two-tailed p-value: 0.877 and 7.69e-54), moderately correlated for non-essential non-PPE genes (Pearson correlation coefficient and two-tailed p-value: 0.464 and 1.12e-40), and weakly correlated for essential genes (Pearson correlation coefficient and two-tailed p-value: 0.222 and 1.51e-6)". We also agree with the reviewer's statement that indel rates in genes are an indication of potential essentiality, and showing that for *M. tuberculosis* was our motivation behind this analysis and figure.

Critique: Minor Comments:

While I find this a very interesting study, in a number of places I would prefer the authors to overstate the evidence for selection. As one example, in the Abstract, I might suggest "Nineteen scars showed evidence of consistent with (or suggestive of) convergent evolution. Our results suggest that sequentially occurring indels are may be an evolutionary mechanism for short-term gene silencing and adaptation in *M. tuberculosis*." These kinds of statements appear through the manuscript.

Response: In response to the reviewer's suggestion, we have edited this statement as follows: "Based on phylogenetic analysis, eight scars evolved in multiple clinical isolates likely by convergent evolution, although scars descending from an evolutionary ancestor were more common".

Critique: Phylogeny analysis. As mentioned, I would have preferred that all of the analyses were carried out against a fixed phylogeny (downsampling for display purposes, but always in the same way). For example in Figures S6 to S9, it is confusing to try to compare the analyses in different panels. Much of the time one can map the phylogenies to each other by eye, but sometimes the phylogenies in some subpanels look quite different, presumably due to the sampling. Moreover, I would have liked to see more analysis showing the relationship of the two (or more) indels forming a scar. The authors indicate that when a pair of indels form a scar event, one (or sometimes both?) of these indels is sometimes seen by itself in other samples, as would be expected. One might expect to observe a number of cases where the first ("wounding") indel occurs near the root of the tree and then second ("fixing") indel occurs one or more times closer to the leaves of the tree. The branch lengths in the phylogeny would provide information about the timing of these indels and perhaps additional evidence about whether there is any selective pressure to "correct" a frame shift in a particular gene.

Response: (i) The relevant figures are Fig. S7b-d in the revised version. The phylogenies look different because each tree has isolates with scars in the gene + same set of randomly selected 200 isolates that are non-scarred and phylogenetically distant. We earlier tried to include all 414 scarred isolates + 200 random non-scarred isolates, but that was too much for the tree drawing programs that can show individual isolates in different colors (as we wished to do).

(ii) We have now included Figure S6, that shows phylogenetic relationships of scar indels seen in two of the ten scars in *espl* gene. The panel (a) in this figure specifically shows that both indels that form a scar may appear independently. Panel (b) shows a case where both indels are common in a clade but not outside of it. We also show in Figure S7a and c where (much as the reviewer predicted) several isolates share one of the same indels (presumably but not proven to be the wounding indel), but have a different fixing indel.

Critique: Figure 6. I like Figure 6 quite a lot, and the phylogenetic analysis in general. However, I have a number of suggested improvements: a) Rather than identifying the scars by number (1-10) on the Y axis of panel (b), I would suggest giving them letters A-J. b) I find it hard to match up the events on panel (a) with panel (b) by matching the colors (where is the yellow event that is visible on panel (a)?). I would suggest that in addition to using color, label the events on panel (a) with the letters (A-J) of the scar on panel (b) near the event in order to make the correspondence unambiguous. You could also refer to these labels in the text (instead of referring to the scars by color), which would make the work more accessible to people with limited color perception. c) I want to know the frequency of the events more clearly. It is clear that the gray event is of much higher frequency and has expanded in a particular clade. Perhaps you could write the number of strains where the scar was detected to the right of each scar on panel (b). Even for the less frequent events, I would like to know if they are singletons, doubletons, etc. d) It is not obvious when two indel sites are shared or are simply close by. Perhaps this could be indicated somehow visually (perhaps a vertical dashed line?). e) Assuming the last two indels are the same, the purple and red events (numbers 2 and 9) look likely to be derived from the gray event (number 5). I couldn't tell if this matched the phylogeny. f) If the last two indels in scars number 2, 5 and 9 are the same, then since the gray event restores frame, the middle segment of the purple event must also be in frame. This suggests that the purple event is actually two small scars, not one long one. And the same for the red event, although the situation here might be more complicated. Two short scars also increase the evidence that the scars in this gene may be under selective pressure to be short.

Response: We thank the reviewer for making thoughtful suggestions to improve Figure 6.

- a) We have now also indicated scars by letters, as suggested by the reviewer.
- b) We apologize for figure not being easy to interpret. We had used yellow to denote a scar in both panels but realized that yellow was difficult to see in panel b, and changed it to purple, but forgot to change the color in the tree. We have now fixed this error.
- c) We have now added the frequency of each scar in the panel (b).
- d) We have added dashed lines to indicate the genic positions where scar indels occur.

e) The gray, red, and purple (earlier seen as yellow in the tree) all fall in the same clade in the tree. Thus, phylogeny does support the evolutionary relatedness of these scars.

f) The panel (b) shows scar indels as circles, and the lines connected them was simply used to aid in visualization of all indels belonging to a scar as a set. It does appear that the purple event is composed of two scars. The line is simply to indicate that all four indels were found together in at least one isolate. The red scar event, however, has one scar same as the gray event, but the other scar composed of three indel events that are not as close to each other as other scar events in this gene. To explore if scars are under selective pressure to be short, we had plotted distance between two scar indels (figures 5a and 5b) that showed that indeed, the scar indels occur close to each other.

Critique: Regarding Figure S2, comparing indel calls between tools is notoriously difficult, due to different callers choosing slightly different representations for the indel. The authors didn't mention how they handled this. Did you use a normalization tool? Did you use a less restrictive test than an exact allelic match?

Response: We apologize for omitting this detail in our earlier version. We have now included section "Reformatting indel calls from SAMtools and GATK HaplotypeCaller" in online methods that states that indels were reformatted as "<ref_base>_<genomic_position>_<type_and_indel_len><indel_string>", where ref_base is the reference genome base at the genomic position where indel was detected, indel "type" is "+" for insertion and "-" for deletion, and indel_string is the bases that were inserted or deleted". Exact matches of these indel representations were done when comparing indel calling tools.

Critique: In the methods section "Evaluation of indel and scar detection by ScarTrek". Near the end it says "5.2% of indels detected by ScarTrek were not found by SAMtools and GATK HaplotypeCaller". You should say "SAMtools and/or GATK", as you do in the main text, and be consistent with the language in the rest of that paragraph.

Response: We thank the reviewer for catching this error. This has now been fixed.

Critique: In this same section: At the end, the text claims ScarTrek has 86.7% accuracy. I was unable to discern how you derived this number. I would also avoid using the word "accuracy" based on comparisons between callers. If this is derived somehow from the number of overlapping calls, I would just explain how you derived it.

Response: The sentence claimed that ScarTrek "detected scars with 86.7% accuracy". Since in the last version 85 scars out of 98 predicted scars turned out to be correct, the accuracy in scar detection is $(85/98) \times 100 = 86.7\%$. However, we have now deleted this sentence because we realize that this is not a true measure of accuracy because we did not start with a known set of scars against which ScarTrek's performance could be compared.

Critique: Figure 5a does not appear to be referenced from the main text.

Response: We apologize for this error. Fig 5a is now referenced in the manuscript as follows: "We found that the indels constituting a scar often occurred close to each other, limiting the number of residues that change in the restored protein (Fig. 5a)."

Critique: Line 97: Says "frame-shifting indels were uniformly distributed along the gene length for the PE-PPE and 98 non-essential genes (Fig. 1c)". However according to the figure description, Figure 1c shows all indels, not just frame-shifting indels. So either the legend is wrong or Fig 1c does not speak directly to this statement.

Response: We apologize for this error. The figure 1c shows all indels (not just frame-shift indels) and we have edited the text to reflect that.

Critique: Figure 3: Could use a legend to indicate the meaning of the colors rather than just putting this in the figure description.

Response: Legend has been added to Fig. 3 and every other scatter plot in the manuscript.

Critique: I really liked supplementary Figure S3.

Response:

Thank you! Although most of the figures in this manuscript and SI are generated by python scripts, the figure S3 was generated using R.

Reviewers' Comments:

Reviewer #5:

Remarks to the Author:

The authors have addressed my comments from the previous review regarding indels in low complexity regions and the thresholds used for variant calling and contamination. However, I still have some suggestions for the phylogenetic analysis.

The phylogenetic approach used here (neighbor joining based on SNP distances) is not standard in the field for *M. tuberculosis* (or other microbial) genomics. While neighbor joining has been used in a few examples presented by the authors (although the only example with whole genome sequencing data also used Maximum Likelihood and Bayesian methods), there are also numerous examples of phylogenetic analysis of hundreds to thousands of isolates using maximum likelihood or Bayesian approaches (Farhat et al. 2013 *Nature Genetics*, Manson et al. 2017 *Nature Genetics*, Coll et al. 2018 *Nature Genetics*). If neighbor joining is used, distances should be estimated using an evolutionary model instead of a SNP distance matrix.

Here's a nice blog post on performing phylogenetic inference with IQTree as an example:

<https://bitsandbugs.org/2019/11/06/two-easy-ways-to-run-iq-tree-with-the-correct-number-of-constant-sites/>

Additionally, there are phylogeny visualization tools that are compatible with large datasets including ITOL and Microreact (these also have the benefit of being interactive if readers want to explore the data). The phylogenetic analysis would be easier for the reader to follow if all isolates were included (or the dataset was subsampled to remove identical/extremely closely related isolates) instead of multiple subsamples to 200 isolates that are different for each gene with evidence of scars.

A scale for branch lengths should be included in visualizations (for example, Figure 6), and a description of how the trees are rooted should also be included in the methods or the figure legends. These visualizations could also be improved by additionally showing the presence of the wounding indels. However, I realize there are numerous wounding indels, and this may not be feasible to display.

Minor comments:

Line 78: I would remove the word "significantly" from this sentence since these are not the results of a significance test.

Line 95-96: For these eleven genes, does this mean that the annotated start codon is one of the alternative start codons (GTG/TTG)? Could you clarify this in the text?

Line 211-212: Since there are not that many, could you list all genes with evidence of convergent repair of frameshifting indels here instead of only in the supplement?

Reviewer #6:

Remarks to the Author:

The authors largely addressed my concerns on the previous version of the manuscript.

I would like to note in particular that the addition of Figure S2b provides reassurance that false indel calls are likely not contributing unduly to the observations of the scars noted by the authors.

There are two things that I would like to address, however:

1. The addition of supplementary figures S10a-e, in response to my comments, are very nice. The analysis I really had in mind, though, was to combine these figures S10a-e, which show the overall distributions, with similar plots that show the distribution of the same metrics for the indel calls at the 157 (formerly 188) scar-forming indel sites. It would be reassuring to see that these indel calls are not

outliers on these metrics (or if they are outliers, that they are, say, enriched for higher mapping quality, not lower, etc.).

Similarly, for Figure S9a-b, it would be reassuring to see a similar analysis of whether the isolates where scars were detected are outliers in these distributions of isolate-level properties. So, for example, one would hope that it is not the case that the isolates in which scars were detected are all low coverage.

Given that the other quality analyses have not revealed substantial problems, I don't expect this to be the case, but this is the kind of additional analysis I had in mind in my original comments.

2. The addition of Table S10, showing the length distribution of indels > 5bp in response to my comments, is very reassuring about indel quality in my view. There is a clear excess of both insertions and deletions at lengths of +/- 6 up to +/- 18! In each case, the number of indels at length $k=3*n$ is greater than at length $k-1$ and $k+1$. This would be expected, because while an in-frame insertion/deletions will alter protein function (and reduce fitness) a frame-shifting indel should, on average, reduce fitness more than a in-frame indel.

In my view, the authors should not have added this statement on line 78: "In-frame indels were not significantly abundant likely because, unlike synonymous base substitutions, in-frame indels still alter translated protein sequences (add or remove amino acids), and thus may have an appreciable fitness cost." There are two competing forces here: The first is that shorter indels occur, on average, more commonly than longer indels. The second is that in-frame indels should be observed, on average, more than expected based on their length. This is clearly true for the indels 5bp+, which is good. Reading the graph, I think it is also true for the +/- 3bp indels that they are enriched compared to the empirical trend based on length, but it is not as clear. I do suspect that this may be due, in part, to a higher rate of false calls among the 1bp and 2bp indels compared to longer indels, but this is just speculation and it is hard to know for sure without doing something like experimental validation. Overall, however, this analysis suggests reasonably good call quality and rather than the statement the authors added, I would have gone with emphasizing that the in-frame indels were observed overall to be enriched over expectation.

Reviewer #5 (Remarks to the Author):

Critique 1. The authors have addressed my comments from the previous review regarding indels in low complexity regions and the thresholds used for variant calling and contamination.

Response: We thank the reviewer for the helpful suggestions in the previous review on these topics, and for finding our responses satisfactory.

Critique 2. However, I still have some suggestions for the phylogenetic analysis. The phylogenetic approach used here (neighbor joining based on SNP distances) is not standard in the field for *M. tuberculosis* (or other microbial) genomics. While neighbor joining has been used in a few examples presented by the authors (although the only example with whole genome sequencing data also used Maximum Likelihood and Bayesian methods), there are also numerous examples of phylogenetic analysis of hundreds to thousands of isolates using maximum likelihood or Bayesian approaches (Farhat et al. 2013 Nature Genetics, Manson et al. 2017 Nature Genetics, Coll et al. 2018 Nature Genetics). If neighbor joining is used, distances should be estimated using an evolutionary model instead of a SNP distance matrix.

Here's a nice blog post on performing phylogenetic inference with IQTree as an example: <https://bitsandbugs.org/2019/11/06/two-easy-ways-to-run-iq-tree-with-the-correct-number-of-constant-sites/>

Response: We thank the reviewer for this insightful advice. In this revision, we now confirm all phylogenetic analysis by the Bayesian method using the program MrBayes as described in Farhat et al 2013 Nature Genetics paper (suggested by the reviewer). In addition to using the GTR model and a stop criterion of a standard deviation of split frequencies of <0.05 , we ran the MCMC simulations longer (till 300,000 generations) if the convergence diagnostic PSRF for all parameters did not approach 1 (if values were >1.15 or <0.85). Running the simulations for longer resulted in lower split frequencies as well (average split frequencies from all replicates in convergent evolution analysis: 0.02). For the convergent evolution analysis (Fig S7), phylogenies generated via Bayesian approaches were in agreement with NJ method for all but one result: the two scars in gene *eccE1* no longer showed convergent evolution (Fig S7b). This has been pointed out in the legend of the Fig S7b, and in the main manuscript text, where we no longer consider scars in *eccE1* as showing convergent evolution. For the homoplasy analysis (*i.e.*, enumerating incidence of homoplasy in indels in high vs. low complexity regions), the results from Bayesian analysis were in

agreement with those from NJ analysis, however, the MCMC simulations for Bayesian approach failed to converge for one low complexity indel (simulations were run for >1,000,000 generations) and one high complexity indel (simulations were run for 500,000 generations). Unfortunately, this high complexity indel was one of the four high complexity indels that had shown homoplasy (Fig S8a). Thus, we could not confirm homoplasy in this indel with Bayesian approaches, and this has been indicated in the legend of Figure S8b, that shows the NJ trees for the four high complexity indels showing homoplasy. Our final result reported from this analysis was that indels in low-complexity regions showed homoplasy three times as often as high-complexity regions, and this result remained unchanged.

Critique 3. Additionally, there are phylogeny visualization tools that are compatible with large datasets including iTOL and Microreact (these also have the benefit of being interactive if readers want to explore the data). The phylogenetic analysis would be easier for the reader to follow if all isolates were included (or the dataset was subsampled to remove identical/extremely closely related isolates) instead of multiple subsamples to 200 isolates that are different for each gene with evidence of scars.

Response:

We had created a neighbor-joining tree with the entire dataset [we used the NJ method here due to faster computation time compared to any other phylogenetic method (hours compared to days or weeks for faster ML or Bayesian approaches). Other approaches such as FastTree (used in Manson et al. 2017 Nature Genetics) also start with a heuristic NJ tree], however we felt that the ~6,000 isolates in the dataset was too much to display on one tree: each branch had so many isolates that it was impossible to see the evolutionary relationships between the few scarred isolates even without any text/colored nodes on the tree to identify such isolates. Thus, we decided on sampling a smaller set of non-scarred isolates after removing closely related isolates in the entire dataset (588 non-scarred phylogenetically distant isolates selected from the entire dataset). Further, for each replicate, we had used the same 200 independent non-scarred isolates for each gene. We apologize if these details were not clear from the manuscript/methods, we have tried to make this clearer in this revision. The trees look different because we construct a tree with the same 200 non-scarred isolates AND scarred isolates from each gene, and each gene has different number of scarred isolates (thus total number of isolates in each gene tree are different). We have reiterated the procedure here and have done the same in the

“identifying scar clades” section of the methods (in supplement), which we now hope the reviewer finds sufficiently clear.

1. A neighbor-joining tree was constructed of the entire dataset to get an estimate of evolutionary relationships for the sole purpose of selecting phylogenetically distant non-scarred isolates.
2. From this NJ tree of ~6,000 isolates, we selected non-scarred isolates that were separated by at least three nodes in the tree. This yielded 588 isolates, where each isolate was at least three nodes away from every other isolate in this set. There are no identical or closely related isolates in this set. Further details can be found in section “**Identifying phylogenetically distant isolates**” in online methods.
3. We randomly selected 200 isolates from this set of 588 non-scarred phylogenetically distant isolates. [There are 665 isolates that are phylogenetically distant, of which 588 are non-scarred. In the previous version, we had mistakenly mentioned 665 isolates as phylogenetically distant *and* non-scarred, this has been corrected in this revision]. To this set of 200 isolates, we added the isolates that were scarred for a particular gene. For example: *espl* gene scars were found in 57 isolates, making the final data set for this gene as 257 isolates, to see the evolutionary relationships of the 57 *espl* scar isolates against a background of 200 non-scarred evolutionarily distant isolates. All the trees shown in the manuscript are from a single replicate and have the same 200 non-scarred isolates, but they look different because of different number of scarred isolates in each gene.
4. Step 3 was repeated 9 more times (thus, a total of 10 replicates) to generate adequate statistics for determining incidence of scars in our dataset.

Critique 4. A scale for branch lengths should be included in visualizations (for example, Figure 6), and a description of how the trees are rooted should also be included in the methods or the figure legends. These visualizations could also be improved by additionally showing the presence of the wounding indels. However, I realize there are numerous wounding indels, and this may not be feasible to display.

Response:

We have now included a scale for the branch lengths in tree visualizations. We apologize for this omission. All the trees are unrooted. Supplementary Figure 6 shows the isolates that had scarred indels for a couple of scars in the gene *espl*,

but there are a lot of wounding indels in all and thus not feasible to show all such evolutionary relationships.

Minor comments:

Line 78: I would remove the word “significantly” from this sentence since these are not the results of a significance test.

Response: We have removed this line from the manuscript.

Line 95-96: For these eleven genes, does this mean that the annotated start codon is one of the alternative start codons (GTG/TTG)? Could you clarify this in the text?

Response:

We apologize for the ambiguity. In this study we focused on ATG start codons only, and have clarified this in the manuscript and the supplement.

Line 211-212: Since there are not that many, could you list all genes with evidence of convergent repair of frameshifting indels here instead of only in the supplement?

Response:

We thank the reviewer for this suggestion. We now list the genes that show convergent repair of frame-shifting indels in the “Convergent evolution of frame-shift scars” section of Results.

Reviewer #6 (Remarks to the Author):

The authors largely addressed my concerns on the previous version of the manuscript.

I would like to note in particular that the addition of Figure S2b provides reassurance that false indel calls are likely not contributing unduly to the observations of the scars noted by the authors.

There are two things that I would like to address, however:

Critique 1. The addition of supplementary figures S10a-e, in response to my comments, are very nice. The analysis I really had in mind, though, was to combine these figures S10a-e, which show the overall distributions, with similar

plots that show the distribution of the same metrics for the indel calls at the 157 (formerly 188) scar-forming indel sites. It would be reassuring to see that these indel calls are not outliers on these metrics (or if they are outliers, that they are, say, enriched for higher mapping quality, not lower, etc.).

Similarly, for Figure S9a-b, it would be reassuring to see a similar analysis of whether the isolates where scars were detected are outliers in these distributions of isolate-level properties. So, for example, one would hope that it is not the case that the isolates in which scars were detected are all low coverage.

Given that the other quality analyses have not revealed substantial problems, I don't expect this to be the case, but this is the kind of additional analysis I had in mind in my original comments.

Response:

We are happy that the reviewer was satisfied with the data quality assessments included in figures S9 and S10. We did understand from the last review that the reviewer was interested in such an analysis specifically for scar indels and scarred isolates, however, we decided to analyze the entire dataset (all indels and isolates) for read mapping and indel calling quality because:

- (i) The first half of the manuscript discusses indel properties in general (Figures 1-3 are not scar related), and thus analyzing the complete dataset was warranted for increasing confidence in those analyses, in addition to the scar analysis. Further, the scar indels are multi-indel events, and are not classified as scar indels unless all indels involved in a scar are present in an isolate. Thus, there is nothing special about the individual scar indels unless they occur together, and we do see a lot of individual wounding indels (without the second “fixing” indel that make the multi-indel event a scar) in the majority of the isolates. It is also not clear if the “orphan” scar indels (when only one of the scar indels are present in an isolate, which is true for majority of isolates, Table S7) should be excluded from analysis of scar indels alone.
- (ii) The isolates/indels that do not meet the quality thresholds were excluded from all the analyses. Thus, the quality of any subset of data would be within the trends shown in figures S9 and S10, which as reviewer also agreed, was acceptable.
- (iii) The scarred isolates and scar indels that passed the quality thresholds were manually inspected for peculiarities, such as scar indels occurring exclusively in reads but not together. A number of scars were discarded after this step.

We think that the above steps give a comprehensive picture of our dataset and indel calling quality. Further, we have also attached a separate file that contains the raw quality data for all indels (>500,000 indels in total in the dataset).

Critique 2. The addition of Table S10, showing the length distribution of indels > 5bp in response to my comments, is very reassuring about indel quality in my view. There is a clear excess of both insertions and deletions at lengths of +/- 6 up to +/- 18! In each case, the number of indels at length $k=3*n$ is greater than at length $k-1$ and $k+1$. This would be expected, because while an in-frame insertion/deletions will alter protein function (and reduce fitness) a frame-shifting indel should, on average, reduce fitness more than an in-frame indel.

In my view, the authors should not have added this statement on line 78: "In-frame indels were not significantly abundant likely because, unlike synonymous base substitutions, in-frame indels still alter translated protein sequences (add or remove amino acids), and thus may have an appreciable fitness cost." There are two competing forces here: The first is that shorter indels occur, on average, more commonly than longer indels. The second is that in-frame indels should be observed, on average, more than expected based on their length. This is clearly true for the indels 5bp+, which is good. Reading the graph, I think it is also true for the +/- 3bp indels that they are enriched compared to the empirical trend based on length, but it is not as clear. I do suspect that this may be due, in part, to a higher rate of false calls among the 1bp and 2bp indels compared to longer indels, but this is just speculation and it is hard to know for sure without doing something like experimental validation. Overall, however, this analysis suggests reasonably good call quality and rather than the statement the authors added, I would have gone with emphasizing that the in-frame indels were observed overall to be enriched over expectation.

Response:

We are thankful to the reviewer for the suggestion of including Table S10 in the supplementary, which shows an uptick in in-frame indels. We agree that the in-frame indels are observed more than expected based on their length, however, in light of no experimental validation or a rigorous statistical analysis to this effect, we have removed this sentence. We'd like to further add that the raw data for all figures is available in the accompanying "Source File", which shows that the +/- 3bp indels are enriched compared to the empirical trend based on length.

Reviewers' Comments:

Reviewer #5:

Remarks to the Author:

The authors have addressed my previous concerns by confirming the Neighbor Joining based phylogenetic results with a Bayesian method and clarifying the subsampling methodology.

I have one additional comment regarding rooting of the trees. While these may be unrooted trees, the display is making them appear rooted because of the defaults of the Newick utilities package.

From the documentation:

"The Newick format is implicitly rooted, in the sense that there is a 'top' node from which all other nodes descend."

The trees would probably be a little easier to interpret and aesthetically pleasing if they were rooted somewhere other than the top node in each individual tree. It doesn't look like midpoint rooting the trees is an option in Newick utilities package. Are there examples from Lineage 7 or the *M. africanum* lineages that could be used as an outgroup clade (as described in the Newick utilities package documentation) to root the tree in the subsampled dataset? If the trees can't be rerooted, clarifying that the trees are meant to be unrooted and the visualization is simply rooted on the top node would be helpful in the figure legends.

I look forward to seeing this work published.

Reviewer #6:

Remarks to the Author:

As I wrote in my original review, since the novel aspects of this paper are the results about scar-forming indels, what seems most important about the quality of the indel calling is whether any of the conclusions about scar-formation are driven by errors in the calling (either false positive sites or inaccurate genotypes). My concern is not so much about reducing the overall error rate in the call set, although that is important, but whether analyzing unusual patterns in scar-forming indels is enriching for any error modes that are present in the call set.

Since the authors declined to test for error enrichment among the scar forming indels, I attempted to use the provided supplementary to do at least some cursory checking myself. This turned out to be difficult to do, however, because the supplementary data uses two different conventions to identify the indel sites. The data supporting supplementary figures 10a-10e are based on reference genome coordinates while the data on the scar-forming indels is based on offsets relative to annotated transcripts with no easy way to convert between the two.

I recommend that in the final version of the manuscript the authors include as supplementary material the full QC'd call set as a (standard) VCF file, including both the indel sites and the genotypes called in each isolate. In addition, the tables referring to the scar forming indels should cross-reference the VCF, either by genomic coordinate or an assigned site ID. This will facilitate reanalysis of the author's conclusions by interested readers.

While attempting to check for error enrichment, I observed that the number of indels called in each isolate varied widely, with a surprising distribution (see attached plot). This does not appear to be

correlated with overall sequencing depth or the other quality metrics reported by the authors. This leads me to wonder whether this distribution is correlated with the source of the sequencing data, which was obtained from three different studies. If so, this would raise a concern about batch effects that could confound the analysis. I apologize to the authors that I did not notice in my original review that they were combining data generated from three separate studies and think to ask how they controlled for batch effects. In any event, this heterogeneity of the indel rates per isolate within the call set should be noted in the supplement and the source of the heterogeneity explained so that interested readers do not have to discover and try to understand it themselves.

Since I was requested to review this manuscript principally with respect to indel calling methodology, I next resorted to more indirect measures to try to inform my judgment on call set quality. Many of the examples of scar-forming indels shown by the authors are compelling and the phylogenetic analyses suggest reasonable biological explanations. But not all do. One example that stands out to me is *sppA* (supplementary figure 7b). First, the phylogenetic tree would suggest an extremely large rate of convergent evolution (ten or more independent events?). Second, the analysis is driven by just two indel sites, at least one of which is observed only in isolates from the Walsh study (supplementary table 6), which heightens my concerns about batch effects. My main concern here is not whether the observed scar found at *sppA* is caused by artifacts in the call set (although that is important). My main concern is whether *sppA* is pointing to an `_error mode_` that may be affecting sites involved in other scar-containing genes as well, potentially at a lower frequency which would make the error mode more difficult to spot in those examples.

In reviewing the Walsh study, I see that they called indels in their data using Cortex. Perhaps the authors could ask the authors of the Walsh study for the Cortex indel calls on those isolates. Comparison of the ScarTrek calls to calls from Cortex, which uses a different calling approach than samtools or GATK, might help shed light on the lingering questions of call set accuracy in this analysis.

We thank the reviewers for their feedback. Please find our point-by-point response below. While updating the manuscript for this revision, we realized that some scar indels were excluded from the final indel set in the last revision after quality thresholds of mapping quality and strand bias were applied, but we didn't update the number of isolates that had those scars. These numbers are now updated in the current version.

Reviewer #5 (Remarks to the Author):

The authors have addressed my previous concerns by confirming the Neighbor Joining based phylogenetic results with a Bayesian method and clarifying the subsampling methodology.

I have one additional comment regarding rooting of the trees. While these may be unrooted trees, the display is making them appear rooted because of the defaults of the Newick utilities package.

From the documentation:

“The Newick format is implicitly rooted, in the sense that there is a ‘top’ node from which all other nodes descend.”

The trees would probably be a little easier to interpret and aesthetically pleasing if they were rooted somewhere other than the top node in each individual tree. It doesn't look like midpoint rooting the trees is an option in Newick utilities package. Are there examples from Lineage 7 or the *M. africanum* lineages that could be used as an outgroup clade (as described in the Newick utilities package documentation) to root the tree in the subsampled dataset? If the trees can't be rerooted, clarifying that the trees are meant to be unrooted and the visualization is simply rooted on the top node would be helpful in the figure legends.

I look forward to seeing this work published.

Response:

We thank the reviewer for suggesting Bayesian analysis and we are happy that our revisions satisfied the reviewer's concerns. We tried another commonly tree drawing program Figtree but did not find a way to color individual leaves. We have thus added the following comment regarding rooting of trees in the “Data Visualization” section of Methods, and in the figure legends: “Although unrooted, the Newick tree visualizations are implicitly rooted at the top node.”

Reviewer #6 (Remarks to the Author):

Critique 1: As I wrote in my original review, since the novel aspects of this paper are the results about scar-forming indels, what seems most important about the quality of the indel calling is whether any of the conclusions about scar-formation

are driven by errors in the calling (either false positive sites or inaccurate genotypes). My concern is not so much about reducing the overall error rate in the call set, although that is important, but whether analyzing unusual patterns in scar-forming indels is enriching for any error modes that are present in the call set.

Since the authors declined to test for error enrichment among the scar forming indels, I attempted to use the provided supplementary to do at least some cursory checking myself. This turned out to be difficult to do, however, because the supplementary data uses two different conventions to identify the indel sites. The data supporting supplementary figures 10a-10e are based on reference genome coordinates while the data on the scar-forming indels is based on offsets relative to annotated transcripts with no easy way to convert between the two.

Response:

We apologize if the reviewer felt that we did not take their comment seriously. We thought we were going a step ahead and doing a more expansive analysis of all the indels and not just the scar indels, since any indel can become a scar indel by occurrence of another indel. It is now clear to us that the reviewer wanted to see if there is anything peculiar about the scar indels alone. To address this concern, we have generated new supplementary figure S11 that shows the quality data for the scar indels. The trends look similar to overall data quality of all indels, which the reviewers have found satisfactory.

We have also updated the supplementary table 6 to show both the transcript as well as the genome coordinates of scar indels, and we sincerely apologize for this oversight. In addition, we have updated the raw data underlying the Supplementary Figure S10 to include a string of the format “scar_indel_<gene_name>” at the end of each line representing an indel, if the indel is a scar indel in gene mentioned in <gene_name>. For example, the following line shows quality data for a scar indel in the gene “sppA”:
ERR550924,815680,C,+1A,63.0,40,0.634920634921,36.5357,33,7,0.175,0.65,scar_indel_sppA

With the genome coordinate and gene name specified for each scar indel, the indel quality information can be cross-referenced with information in Supplementary Table S6.

Critique 2: I recommend that in the final version of the manuscript the authors include as supplementary material the full QC'd call set as a (standard) VCF file, including both the indel sites and the genotypes called in each isolate. In addition, the tables referring to the scar forming indels should cross-reference the VCF, either by genomic coordinate or an assigned site ID. This will facilitate reanalysis of the author's conclusions by interested readers.

Response: We have now added gene name and genome coordinates for all scar indels in the supplementary tables, allowing them to be cross referenced to facilitate reanalysis by interested readers.

It is a standard practice to provide data as a single CSV file (as was also done by the studies we referenced), which we have followed. This allows all the pertinent data (indel site, reference and indel genotype, number of reads, quality data, etc.) to be included in a single file that is easier to analyze and transfer. One VCF file, on the other hand, is generated for each sample (thus, our dataset of ~ 6000 samples would generate ~6000 VCF files). Further, the VCF format is updated periodically and we have quality data (such as strand bias) that is not part of the current VCF format. Due to these reasons, we believe a single CSV file storing all the data in easily accessible comma separated format is the best way to provide supplementary material to the readers.

Critique 3: While attempting to check for error enrichment, I observed that the number of indels called in each isolate varied widely, with a surprising distribution (see attached plot). This does not appear to be correlated with overall sequencing depth or the other quality metrics reported by the authors. This leads me to wonder whether this distribution is correlated with the source of the sequencing data, which was obtained from three different studies. If so, this would raise a concern about batch effects that could confound the analysis. I apologize to the authors that I did not notice in my original review that they were combining data generated from three separate studies and think to ask how they controlled for batch effects. In any event, this heterogeneity of the indel rates per isolate within the call set should be noted in the supplement and the source of the heterogeneity explained so that interested readers do not have to discover and try to understand it themselves.

Response:

We thank the reviewer for pointing out the tri-modal distribution of number of indels/sample in our dataset. There are seven lineages of *M. tuberculosis* that circulate in the world, with each lineage showing strong association with different geographic regions. Each *M. tuberculosis* lineage can be determined by its genome sequence (Brites and Gagneux, 2015, Immunol. Rev 264(1)). We have now determined the lineage for each isolate in our dataset using the program SNP-IT, and plotted the histogram of indel-rate while color-coding each lineage (supplementary figure S14). This new figure clearly shows that the difference in indel rates is attributable to the *M. tuberculosis* lineage of clinical isolates. As we now discuss in the supplementary materials, the indel rate (Fig S14 x-axis) is overlapping for lineages 2 (purple) and 3 (green), two closely related lineages with a most recent common ancestor (Brites and Gagneux, 2015, Immunol. Rev 264(1)), but were distinct from other *M. tuberculosis* lineages. Lineages 1 and 5, which show the largest number of indels represent so called "ancient" lineages in contrast to the "modern" lineages 1, 2 and 3, which all show smaller numbers of indels.

ref: <https://pubmed.ncbi.nlm.nih.gov/25703549/#&gid=article-figures&pid=figure-1-uid-0>

This finding is certainly novel and is likely to generate considerable interest in the scientific community. We are reluctant to speculate on the biological significance of this finding with the data available. However, our results do demonstrate the value of including indel analysis in all future studies of *M. tuberculosis* populations. We are extremely thankful to the reviewer for their analysis of our data which led to this discovery.

We addressed batch effects in our WGS dataset by following stringent QC rules, as is common for WGS data (Tom et al, BMC Bioinformatics 2017), to:

- (1) Exclude samples that have a high degree of “missingness”. Towards this end, we ignored samples with < 50% read mapping rate and those with average depth across genome < 20X. Although we did not specify this earlier, our dataset also has high “genome-wide coverage”, *i.e.*, more than 95% of the genomic sites were mapped with 5 or more reads (see new supplementary figure S9c). Tools for detecting batch effects, such as Indexcov (Pedersen et al, Gigascience 2017), look for inconsistencies in genome-wide coverage. However, our dataset had excluded samples with poor genome-wide coverage and inconsistent read-mapping (see point 3 below).
- (2) Exclude genomic regions that are difficult to characterize: we removed PE-PPE genes, repetitive MIRU-VNTR regions, and homopolymer regions of length 5 nt or more.
- (3) Exclude genomic regions with biased read mapping: we ignored indels in regions where read depth was more than twice the average genomic depth.
- (4) Exclude low quality indels: we ignored indels that were not supported by at least 10 reads, were not supported well by both forward and reverse reads (strand bias < 0.05), and had average mapping quality of < 10. Further, all indel calls that passed the above quality checks were determined to be high or moderate quality by the chi-square test described in Fang et. al, Genome Med 6, 89, 2014.

Some of these quality metrics were added following suggestions by reviewers, for which we are grateful.

Critique 4: Since I was requested to review this manuscript principally with respect to indel calling methodology, I next resorted to more indirect measures to try to inform my judgment on call set quality. Many of the examples of scar-forming indels shown by the authors are compelling and the phylogenetic analyses suggest reasonable biological explanations. But not all do. One example that stands out to me is sppA (supplementary figure 7b).

First, the phylogenetic tree would suggest an extremely large rate of convergent evolution (ten or more independent events?).

Second, the analysis is driven by just two indel sites, at least one of which is observed only in isolates from the Walsh study (supplementary table 6), which heightens my concerns about batch effects.

My main concern here is not whether the observed scar found at *sppA* is caused by artifacts in the call set (although that is important). My main concern is whether *sppA* is pointing to an `_error mode_` that may be affecting sites involved in other scar-containing genes as well, potentially at a lower frequency which would make the error mode more difficult to spot in those examples.

In reviewing the Walsh study, I see that they called indels in their data using Cortex. Perhaps the authors could ask the authors of the Walsh study for the Cortex indel calls on those isolates. Comparison of the ScarTrek calls to calls from Cortex, which uses a different calling approach than samtools or GATK, might help shed light on the lingering questions of call set accuracy in this analysis.

Response:

The goal of this analysis is show that sequential disruption and then restoration of reading frame is a possible mechanism by which the bacterium responds to selection pressures, and to identify candidate genes where this phenomenon may be seen in clinical setting. We agree that a reasonable biological explanation may not be obvious for all candidate genes without further computational and/or experimental investigations. Safi et al, PNAS 2019, for example, experimentally validated this mode of evolution for the *glpK* gene of *M. tuberculosis*.

Further, we believe that we can only comment if convergent evolution is seen or not, and not on the magnitude of convergent evolution, due to the following reasons:

1. Strict quality thresholds may rule out some indels that can potentially be part of a scar. While ScarTrek detected *sppA* scars in 21 isoaltes, GATK detected them in 34 isolates. Further, 42 isolates in the *sppA* tree had *sppA* scar if we removed most quality thresholds, and 59 isolates had at least one of the scar indels and can potentially be scarred with more time allowed for the second scar indel to appear. Since quality thresholds can influence the number of scarred isolates we detect, we cannot accurately determine the rate of convergent evolution from this type of analysis where there is variability in the genomic data (read depth, mapping quality, etc) from sample to sample. These additional scarred and potentially scarred isolates do seem to appear in the vicinity of the original scar isolates in the tree in figure S7 (see figures 1 and 2 at the end of the

- document), but we cannot make strong quantitative assessments about the rate of convergent evolution.
2. The functional significance of the gene *sppA* is not known in clinical settings, which will influence the importance of quickly restoring its reading frame when disrupted. This is unlike the scenario of the ESX-1 gene cluster, which is known to be important for virulence and observation of scars in several of those genes makes biological sense. It would be desirable to know more about the biological role of the scarred genes in *Mtb* infection and biology, but these experimental investigations are out of scope of the present study.

We are a bit confused by the comment that “the analysis is driven by just two indel sites, at least one of which is observed only in isolates from the Walsh study” (we think the reviewer meant “Walker study”). The scar in the *sppA* gene consists of two indels, like scars in several other genes. And *both* scar indels are present in a scarred isolate. While we do see only one of the scar indels being present in several isolates, we do not count those as scar isolates unless all the indels that constitute a scar are present simultaneously in a given isolate.

The Cortex program detects genetic variation by *de novo* assembly, which is a different approach than the programs that detect indels by alignment methods. In our manuscript, we had discussed these two approaches for indel calling: “There are two main approaches for detecting indels from sequencing data: *de novo* assembly and alignment to a reference genome. It can be difficult to identify indels using either approach because indels usually occur in low complexity sequences that may be prone to errors in genome assemblies and read alignments.^{38,39} Long homopolymer runs also introduce errors during the PCR step of sequencing library preparation.^{39,40} While different variant callers have high concordance in SNP calling, the same is not true for indel calling due to a high number of false positives.^{41,42} Assembly based approaches are well suited for detecting large indels that are very likely to be missed by the alignment based approaches.^{39,43} However, direct comparison of indel-calling in human genome by assembly-based and alignment-based approaches showed that alignment-based approaches achieved a much higher precision and recall.⁴⁴ Recent indel callers combine both approaches by re-assembling the indel region after an initial read-alignment.⁴⁵⁻⁴⁷ Even though indel calling remains problematic and downstream filtering of variants is needed to improve accuracy,^{39,40} ScarTrek detects a lower number of false positives and has high concordance with other indel callers by focusing on indels with strong read support.”

To better assess ScarTrek performance, an alignment based method, we had compared it to other alignment based programs [Samtools (Li, Bioinformatics 2011: 3166 citations) and GATK (DePristo et al, Nature Genetics 2011: 8586 citations)] that are much more widely used than Cortex (Iqbal et al, Bioinformatics 2013: 43 citations). All of the *sppA* scars and scar indels detected by ScarTrek were detected by GATK as well (100% agreement). However, Cortex did not

detect either of the two scar indels in the *sppA* gene. Finally, our phylogenetic analysis of scars in clinical *M. tuberculosis* isolates clearly demonstrates that most scars occur in closely related strains which share a recent common ancestor. This finding provides very strong population-based support that (at least in these cases) our identification of scars is accurate (i.e. both sensitive and specific). Thus, both prior work and several different analyses presented in our manuscript each support our analytic approach.

Figure 1. Bayesian tree showing isolates (in red) with the scar in *sppA* gene. These isolates had both the scar indels, when the scar indels are identified with relaxed quality thresholds.

Figure 2. Bayesian tree showing isolates (in red) with *either or both* the scar indels in the *sppA* gene. These isolates had the scar indels, when the scar indels are identified with relaxed quality thresholds.

Reviewers' Comments:

Reviewer #6:

Remarks to the Author:

It was good to learn that the tri-modal distribution of the number of indels per sample could be traced to the major lineages of *M. Tuberculosis*. The fact that the variation in the number of called indels has a clear biological explanation is nice additional evidence of overall indel callset quality. It seems to me that that way the modes of the distributio map to the lineages also makes sense in the context of the phylogeny shown in the Brites and Gagneux manuscript. The indel distribution in the current work would suggest that the reference strain H37Rv most likely comes from Lineage 4, which seems to be consistent with the accepted phylogeny.

I appreciate that the authors have now included supplementary figure S11 to look specifically at quality metrics for the indels involved in scars. I believe this is critically important because any analysis that enriches for "unusual" occurrences (e.g. recurrent scar-forming indels) is likely to also enrich for indel calling error or genotyping error.

Indel calling is notoriously difficult and I would expect some error enrichment. The fraction of reads supporting indels involved in scars appears to be a bit lower overall (the left tail of S11c appears to be somewhat thicker than in S10c) and there is probably a slightly negative shift in the read coverage of the scar-involved indels (S11a vs. S10a). The mapping quality and strand bias distributions on the other hand seem to be more nearly identical. Overall, based on just visual inspection (no statistical tests) I concur with the authors that while there is some expected enrichment, the rate does not appear to be excessive. In particular, the bulk of the scar-involved indels have quality metrics falling into the "good" parts of the distributions, suggesting that is it valid to extrapolate from the quality of the overall call set to this smaller (non-random) sample that underlies all of the downstream analysis.

I do still have some suspicion that the *sppA* results may be deriving from some kind of artifact in the indel calling, but it is hard to know for sure. Relaxing the filters does show more cohesive clustering, although the results for *sppA* are still not as clear and convincing as the other examples with high-frequency scars. Since the gene has unknown function, I accept the author's argument that strong selective pressure could also be a possibility and thus further study would be quite important.

The authors have addressed all of my concerns.

Response to reviewer

Reviewer #6 (Remarks to the Author): It was good to learn that the tri-modal distribution of the number of indels per sample could be traced to the major lineages of *M. Tuberculosis*. The fact that the variation in the number of called indels has a clear biological explanation is nice additional evidence of overall indel callset quality. It seems to me that that way the modes of the distribution map to the lineages also makes sense in the context of the phylogeny shown in the Brites and Gagneux manuscript. The indel distribution in the current work would suggest that the reference strain H37Rv most likely comes from Lineage 4, which seems to be consistent with the accepted phylogeny. I appreciate that the authors have now included supplementary figure S11 to look specifically at quality metrics for the indels involved in scars. I believe this is critically important because any analysis that enriches for "unusual" occurrences (e.g. recurrent scar-forming indels) is likely to also enrich for indel calling error or genotyping error. Indel calling is notoriously difficult and I would expect some error enrichment. The fraction of reads supporting indels involved in scars appears to be a bit lower overall (the left tail of S11c appears to be somewhat thicker than in S10c) and there is probably a slightly negative shift in the read coverage of the scar-involved indels (S11a vs. S10a). The mapping quality and strand bias distributions on the other hand seem to be more nearly identical. Overall, based on just visual inspection (no statistical tests) I concur with the authors that while there is some expected enrichment, the rate does not appear to be excessive. In particular, the bulk of the scar-involved indels have quality metrics falling into the "good" parts of the distributions, suggesting that it is valid to extrapolate from the quality of the overall call set to this smaller (non-random) sample that underlies all of the downstream analysis. I do still have some suspicion that the *sppA* results may be deriving from some kind of artifact in the indel calling, but it is hard to know for sure. Relaxing the filters does show more cohesive clustering, although the results for *sppA* are still not as clear and convincing as the other examples with high-frequency scars. Since the gene has unknown function, I accept the author's argument that strong selective pressure could also be a possibility and thus further study would be quite important. The authors have addressed all of my concerns.

Response:

We thank the reviewer for the very helpful suggestions that significantly improved our manuscript. We are happy that the reviewer was satisfied with our revision.

In response to the reviewer's final comments we have added the following in the Discussion: "Our finding that indel frequency per sample shows a tri-modal distribution according to *M. tuberculosis* lineage (Fig. S 14), with the "modern" lineages 2, 3 and 4 showing fewer indels than the "ancient" lineages 1 and 5 provides further validation for the accuracy of our indel calls".